# Central role of Tim17 in mitochondrial presequence protein translocation

Laura F. Fielden[1,6], Jakob D. Busch[1,6], Sandra G. Merkt[1,2], Iniyan Ganesan[1], Conny Steiert[1], Hanna B. Hasselblatt[1], Jon V. Busto[1], Christophe Wirth[1], Nicole Zufall[1], Sibylle Jungbluth[3], Katja Noll[3], Julia M. Dung[1], Ludmila Butenko[1], Karina von der Malsburg[3], Hans-Georg Koch[1], Carola Hunte[1,4,5], Martin van der Laan[3✉] & Nils Wiedemann[1,4,5✉]

The presequence translocase of the mitochondrial inner membrane (TIM23) represents the major route for the import of nuclear-encoded proteins into mitochondria[1,2]. About 60% of more than 1,000 different mitochondrial proteins are synthesized with amino-terminal targeting signals, termed presequences, which form positively charged amphiphilic α-helices[3,4]. TIM23 sorts the presequence proteins into the inner membrane or matrix. Various views, including regulatory and coupling functions, have been reported on the essential TIM23 subunit Tim17 (refs. 5–7). Here we mapped the interaction of Tim17 with matrix-targeted and inner membrane-sorted preproteins during translocation in the native membrane environment. We show that Tim17 contains conserved negative charges close to the intermembrane space side of the bilayer, which are essential to initiate presequence protein translocation along a distinct transmembrane cavity of Tim17 for both classes of preproteins. The amphiphilic character of mitochondrial presequences directly matches this Tim17-dependent translocation mechanism. This mechanism permits direct lateral release of transmembrane segments of inner membrane-sorted precursors into the inner membrane.

Presequence-carrying preproteins imported from the cytosol must first traverse the channel of the translocase of the outer membrane (TOM complex) and are then translocated by the translocase of the mitochondrial inner membrane (TIM23) complex across the hydrophobic lipid bilayer of the inner membrane. Until now, Tim23 was thought to function in membrane translocation in the form of a hydrophilic channel[8–12]. However, the role of the additional homologous essential TIM23 subunit Tim17 in membrane translocation of preproteins has not been clarified[1,2,5,6]. TIM23 subunits crucial for translocation across the inner membrane should be in close vicinity of preprotein segments spanning the core of the membrane. Therefore, we accumulated preproteins in the mitochondrial TOM–TIM23 import site and performed site-specific crosslinking. We used two model preproteins of identical length, both consisting of the N-terminal matrix-targeting signal of cytochrome $b_2$ and the carboxy-terminal passenger protein dihydrofolate reductase (DHFR) (Fig. 1a)[13,14]. The mature domain of the preprotein $b_2(84)_{+7}$-DHFR (named $b_2(84)$) contains a transmembrane segment (also called the stop-transfer signal) that is laterally released into the inner membrane. In the preprotein $b_2(110)_{\Delta 19}$-DHFR (named $b_2(110)_\Delta$), the stop-transfer signal has been deleted by removing the residues 47–65, and thus, the preprotein is directed into the matrix[15,16]. Upon incubation of the [35]S-labelled preproteins with isolated yeast (*Saccharomyces cerevisiae*) mitochondria in the presence of the DHFR ligand methotrexate (MTX), the $b_2$ portion enters the mitochondrial

TOM–TIM23 import site. The preprotein is proteolytically processed between residues 31 and 32 by the matrix processing peptidase (MPP), whereas the MTX-stabilized DHFR moiety remains on the cytosolic side of TOM[17–19] (Fig. 1b).

## Precursor proteins interact with Tim17

We generated a C86S cysteine-free (CF) variant (#) of the mature part of $b_2(110)_{\Delta 19}$-DHFR[#] to introduce unique cysteine residues at positions 47–54 of the $b_2(84)$-DHFR and $b_2(110)_\Delta$-DHFR[#] constructs for cysteine-specific crosslinking. These residues mark the start of the transmembrane domain (lateral sorting signal) of the $b_2(84)$-DHFR construct and span the inner membrane when the two precursor proteins are arrested at the TOM–TIM23 import site[14] (Fig. 1b). MPP processing of arrested $b_2(84)$-DHFR constructs (containing the sorting signal) generates the naturally CF wild-type intermediate (i) form of the precursor. Chemical crosslinking with a short (7.3 Å) cysteine- and amino-reactive crosslinker (*m*-maleimidobenzoyl-*N*-hydroxysuccinimide ester (MBS)) revealed efficient crosslinking products with a molecular mass of approximately 45 kDa, specific for cysteines introduced at positions 48–54 (Fig. 1c). With the equivalent matrix-destined $b_2(110)_\Delta$-DHFR[#] protein (lacking the sorting signal), crosslinking products with a molecular mass of approximately 45 kDa were formed for all variants analysed even in the absence of engineered cysteine residues (Extended Data

[1]Institute of Biochemistry and Molecular Biology, ZBMZ, Faculty of Medicine, University of Freiburg, Freiburg, Germany. [2]Faculty of Biology, University of Freiburg, Freiburg, Germany. [3]Medical Biochemistry and Molecular Biology, Center for Molecular Signaling, PZMS, Saarland University, Homburg, Germany. [4]CIBSS—Centre for Integrative Biological Signalling Studies, University of Freiburg, Freiburg, Germany. [5]BIOSS—Centre for Biological Signalling Studies, University of Freiburg, Freiburg, Germany. [6]These authors contributed equally: Laura F. Fielden, Jakob D. Busch. ✉e-mail: martin.van-der-laan@uks.eu; nils.wiedemann@biochemie.uni-freiburg.de

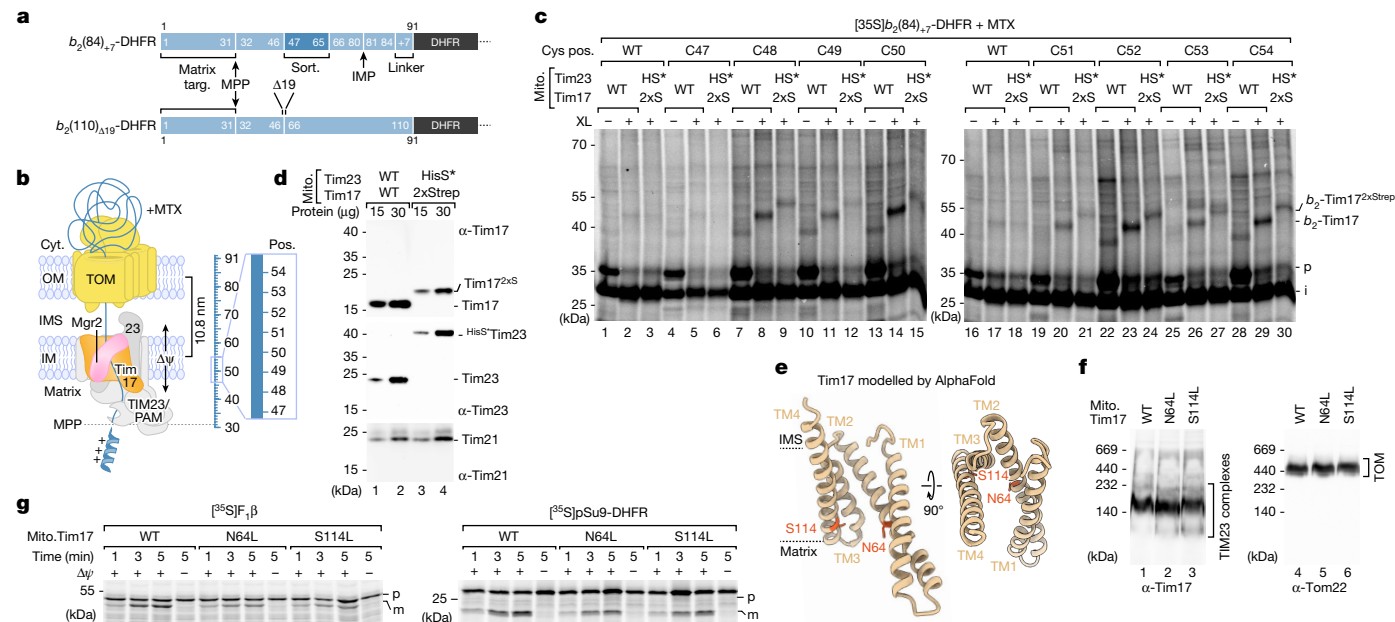

**Fig. 1 | Matrix-targeted and inner membrane-sorted precursors interact with Tim17. a**, Schematic of preprotein constructs of identical length. Both preproteins are derivatives of cytochrome $b_2$ (containing an N-terminal matrix-targeting signal (Matrix targ.)) fused to DHFR. The $b_2(84)_{+7}$-DHFR contains an inner membrane sorting signal (Sort.) containing a transmembrane domain and a seven amino acid linker (+7). The $b_2(110)_{\Delta19}$-DHFR is imported into the mitochondrial matrix because the sorting signal (19 amino acids) is deleted ($\Delta$). IMP, inner membrane protease. **b**, Model of arrested $b_2$-DHFR precursor proteins depicted in **a**. Cyt., cytosol; IM, inner membrane; IMS, intermembrane space; OM, outer membrane; Pos., position; TIM23, presequence translocase of the inner membrane; PAM, presequence translocase-associated motor; $\Delta\psi$, membrane potential. **c**, Import of $^{35}$S-labelled $b_2(84)_{+7}$-DHFR with cysteine residues at the indicated positions (pos.) in wild-type (WT) or Tim17$^{2xStrep}$ (2xS) and $^{HisSUMOstar}$Tim23 (HS*) mitochondria in the presence of MTX followed by chemical crosslinking with MBS (XL). Samples were analysed by SDS–PAGE and autoradiography. $b_2$-Tim17, $b_2(84)_{+7}$-DHFR-Tim17 crosslinking product; i, intermediate; p, precursor. **d**, Immunodecoration depicting the mass shift of tagged Tim17$^{2xStrep}$ and $^{HisSUMOstar}$Tim23. **e**, AlphaFold model of full-length Tim17 (*Saccharomyces cerevisiae*; AF-P39515). Model shown in ribbon representation is viewed parallel to the membrane (left) and from the IMS (right). Dashed lines indicate the predicted membrane region. Orange, asparagine 64 (N64), serine 114 (S114) residues. **f**, Protein complexes of isolated wild-type Tim17 and mutant yeast mitochondria were analysed by Blue Native (BN)–PAGE and immunodecoration. **g**, Import of radiolabelled matrix precursor proteins pSu9-DHFR (pSu9, presequence of *Neurospora crassa* ATP synthase subunit 9) and ATP synthase subunit 2 (Atp2), also named $F_1\beta$, into isolated wild-type Tim17 and mutant yeast mitochondria followed by SDS–PAGE and autoradiography. m, mature.

Fig. 1a). This result suggests that the protein crosslinked to the arrested mitochondrial precursor protein contains itself a cysteine residue, which can be utilized for crosslinking to $b_2(110)_{\Delta}$-DHFR$^{\#}$. Therefore, both experiments were repeated with a short homo-bifunctional cysteine crosslinker (bismaleimidoethane (BMOE)). Position-specific crosslinking products of 45 kDa were generated with $b_2(84)$-DHFR with cysteines from position 48 to 52 and with $b_2(110)_{\Delta}$-DHFR$^{\#}$ for positions 47–50 (Extended Data Fig. 1b,c). The mass of all these observed crosslinking products corresponds to the MPP processed intermediate (i) forms of the precursor proteins (approximately 28 kDa) and the TIM23 core subunit Tim17 (17 kDa). To determine whether the crosslinks really represent an adduct between the precursor and one of the two central subunits of the TIM23 complex, we generated a yeast strain expressing both Tim17 with a 2xStrep tag (Tim17$^{2xStrep}$, approximately 4 kDa molecular mass shift) and Tim23 with a HisSUMOstar tag ($^{HisS*}$Tim23, approximately 14 kDa molecular mass shift)[20] (Fig. 1d). Arrest of the precursor proteins in wild-type Tim17 and in Tim17$^{2xStrep}$(2xS)/$^{HisS*}$Tim23(HS*) mitochondria followed by chemical crosslinking revealed a small specific shift to a higher molecular mass of all crosslinking bands, which corresponds to the increased mass of 2xStrep-tagged Tim17 (Fig. 1c and Extended Data Fig. 1a–c). Together, the efficient Tim17 crosslinking products of $b_2(84)$-DHFR and $b_2(110)_{\Delta}$-DHFR$^{\#}$ between positions 47 and 54 demonstrate that Tim17 is in close proximity to both a matrix-targeted as well as a laterally sorted preprotein accumulated in the TOM–TIM23 import sites (Fig. 1b). To determine whether the crosslinking efficiency to Tim17 is correlated with Tim17 activity, we

arrested matrix-destined and inner membrane-sorted precursor proteins in temperature-sensitive *tim17*-mutant mitochondria after heat shock and performed subsequent chemical crosslinking[21]. The strong reduction in the formation of the precursor protein–Tim17 crosslink in *tim17-4* and *tim17-5* temperature-sensitive mitochondria indicates that crosslinking of precursor protein to Tim17 directly correlates with the activity of Tim17 (Extended Data Fig. 1d). Together, these results indicate that Tim17 is a major interaction partner of mitochondrial precursor proteins translocating the inner membrane.

## Tim17 lateral cavity for translocation

In addition to Tim17 and Tim23, the Tim17 protein family also contains Tim22 (ref. 22). To further analyse the function of Tim17, we modelled the Tim17 transmembrane structure using the cryoelectron microscopy (cryo-EM) structures of Tim22 from yeast and human as templates. Both models are in good agreement with the AlphaFold prediction of Tim17 (Fig. 1e and Extended Data Fig. 2a)[23–25]. However, in contrast to the previous view on mitochondrial protein translocation across the inner membrane, neither the presequence translocase subunits Tim17 and Tim23 nor the carrier translocase (TIM22 complex) subunit Tim22 seem to form channels for precursor protein translocation across or membrane insertion into the inner membrane. The Tim17 and Tim23 models feature four transmembrane domains that form a curved surface, with the lateral cavity opening to the lipid bilayer similar to Tim22[24,25]. To test whether hydrophilic residues within the

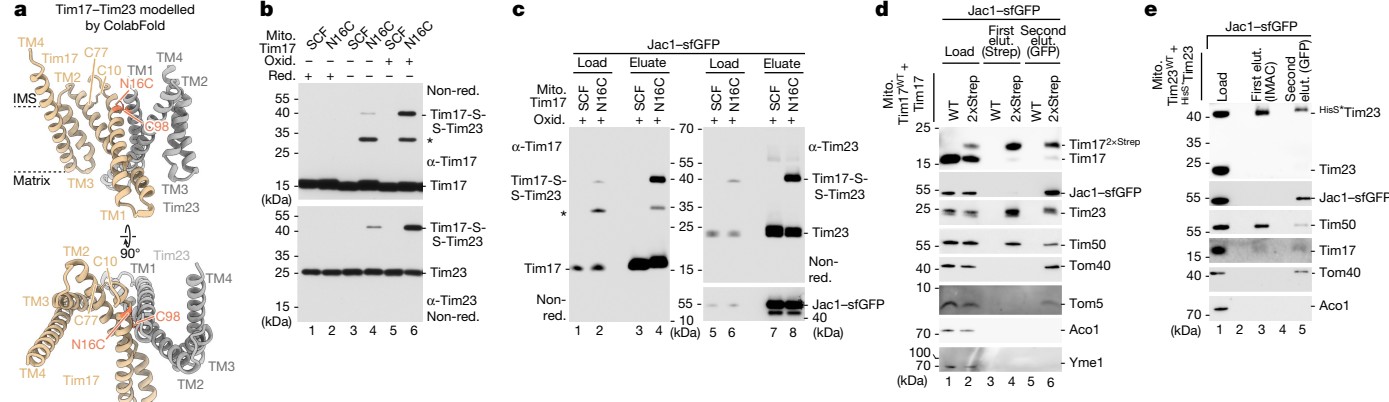

**Fig. 2 | Lateral transmembrane cavities face toward opposite sides of the Tim17–Tim23 dimer. a**, ColabFold structural protein complex model of Tim17 (tan) and Tim23 (grey) heterodimers. Tan, endogenous intramolecular disulfide bond formed between Tim17 Cys10 (C10) and Cys77 (C77); orange, Tim17 Cys16 (N16C) and Tim23 Cys98 (C98) forming an intermolecular disulfide bond. **b**, In organello disulfide bond formation between Tim17 and Tim23. Mitochondria isolated from yeast strains expressing SCF Tim17 or Tim17 N16C were left untreated or treated with the reductant dithiothreitol (DTT; red) or oxidant 4,4′-dipyridyl disulfide (4-DPS; Oxid.) followed by SDS–PAGE. Tim17-S-S-Tim23, disulfide-linked Tim17 and Tim23. *Unidentified Tim17-specific conjugate.

**c**, Import of Jac1–sfGFP into SCF Tim17 or N16C mitochondria followed by oxidation with 4-DPS. Jac1–sfGFP-containing TOM–TIM23 supercomplexes were isolated with a GFP nanobody. Load (1%) and eluate (100%) fractions were analysed by SDS–PAGE and immunodecoration. **d,e**, Import of Jac1–sfGFP into wild-type or Tim17[2xStrep] (**d**) and [HisSumo*]Tim23 mitochondria (**e**). TIM23 complexes were isolated by tandem purification first using streptavidin (Strep; **d**) or immobilized metal affinity chromatography (IMAC; **e**) followed by purification of Jac1–sfGFP-containing TOM–TIM23 supercomplexes using a GFP nanobody. Fractions were analysed by SDS–PAGE and immunodecoration. Load, 0.25%; first elut., first elution (2.5%); second elut., second elution (100%).

lateral cavity of Tim17 are crucial for precursor protein translocation, we generated Tim17[N64L] (the second transmembrane domain, TM2) and Tim17[S114L] (TM4) mutants located on opposing sides of the cavity (Fig. 1e). Compared with wild type, the growth of *TIM17*[N64L] yeast was slightly affected at elevated temperatures (Extended Data Fig. 2b). Isolated Tim17[N64L] and Tim17[S114L] mutant mitochondria from cells grown at low temperature have protein levels, TIM23 complexes and a membrane potential ($\Delta\psi$) across the inner membrane comparable with wild-type protein (Fig. 1f and Extended Data Fig. 2c,d). In contrast to normal processing of mitochondrial presequence proteins sorted into the inner membrane and Dic1 assembly (which is dependent on the TIM22 complex/carrier translocase), the Tim17[N64L] and Tim17[S114L] mutants specifically exhibited strong import defects of radiolabelled presequence proteins destined to the mitochondrial matrix (as assessed by MPP cleavage and formation of the processed intermediate (i) and mature (m) bands) (Fig. 1g and Extended Data Fig. 2e,f). These results indicate that the hydrophilic residues on the matrix side of the lateral cavity of Tim17 are crucial for mitochondrial matrix protein translocation across the inner membrane.

## Tim17 and Tim23 interact back to back

During our study, a structural model of the Tim17–Tim23 core complex was generated based on co-evolution analysis, which is consistent with the ColabFold prediction and a preprint of a TIM23 core complex cryo-EM structure (Fig. 2a and Extended Data Fig. 3a,b)[26,27]. Even though this Tim17–Tim23 heterodimer does not form a channel, its interaction interface is consistent with previous studies. TM1 and TM2 of both Tim17 and Tim23 were found to be crucial for the structural integrity/functionality of the Tim17–Tim23 complex, and therefore, both subunits are essential for viability[28–30]. To prove that Tim17 and Tim23 interact back to back in the native lipid environment, we generated a semi-cysteine-free (SCF) Tim17[C118G_C120V] construct (which only contains two cysteines at positions 10 and 77 required for intramolecular disulfide bond formation)[31,32]. To probe the interaction between Tim17 and Tim23, we introduced specific single-cysteine residues into the SCF Tim17[SCF] (Extended Data Fig. 3c). We engineered the

Tim17[N16C] in close proximity to the natural cysteine 98 of Tim23 located within the first transmembrane domain. On non-reducing SDS gels, a Tim17[N16C]-specific conjugate of approximately 40 kDa was detected by antibodies directed against Tim17 and Tim23, which was absent on reducing gels (Extended Data Fig. 3d). These results indicate that a disulfide bond was formed between Tim17 and Tim23 in organello. Subsequently, we treated isolated mitochondria with an oxidant resulting in a very efficient disulfide bond formation between Tim17 and Tim23, providing evidence that the cavities of Tim17 and Tim23 are indeed not facing each other (Fig. 2b). Specific disulfide bond and cysteine crosslink formation between further cysteine pairs in the Tim17[SCF] and CF Tim23[CF] background further confirmed this back-to-back orientation (Extended Data Fig. 3e,f). To analyse whether the TIM23 complex adopts the same orientation while engaging with a precursor protein, we arrested the recombinant matrix-destined Jac1–sfGFP precursor in Tim17[N16C] mitochondria and performed a subsequent oxidation assay[20]. Analysis of the anti-GFP purification eluate by non-reducing SDS–PAGE shows a similar efficiency of disulfide bond formation between Tim17 and Tim23 compared with the TIM23 complex without precursor (Fig. 2c). This indicates that the TIM23 complex engaged with precursor protein also adopts the back-to-back orientation of Tim17 and Tim23. Another possibility would be that two heterodimeric TIM23 complexes associate in a way that a channel-like structure is formed between two lateral cavities facing each other (for example, the Tim17 lateral cavity of the first heterodimer and the Tim23 lateral cavity of the second heterodimer)[20,33,34]. To analyse the oligomeric state of the Tim17–Tim23 heterodimer during precursor translocation, we arrested recombinant Jac1–sfGFP in mitochondria co-expressing both Tim17[WT] and Tim17[2xStrep] or Tim23[WT] and [HisSUMOstar]Tim23 (Fig. 2d,e, load)[20]. Analysis of the TIM23 complexes purified by Tim17[2xStrep] and [HisSUMOstar]Tim23 reveals a very strong enrichment of the tagged form over the non-tagged form of the same subunit (Fig. 2d,e, first elut.)[20]. To specifically analyse the oligomeric state of the precursor protein-carrying TIM23 complex, a subsequent second anti-GFP purification was performed, which enriches the precursor arrested at the TOM–TIM23 supercomplex (Fig. 2d,e, second elut.). Although the ratio of the tagged Tim17 or Tim23 subunits to the non-tagged

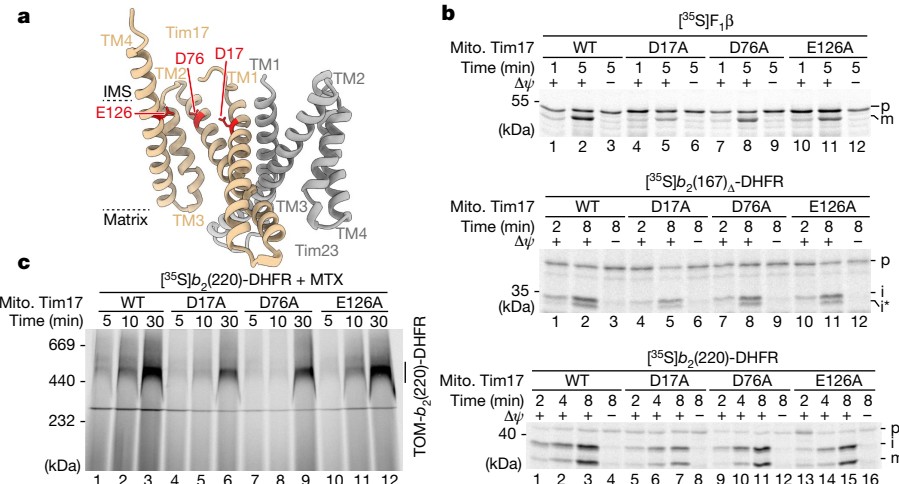

**Fig. 3 | Negative charges of Tim17 transmembrane domains are crucial for presequence protein translocation. a**, ColabFold model of *S. cerevisiae* Tim17 (tan) and Tim23 (grey) heterodimer showing, as red sticks, the locations of negatively charged residues aspartic acid 17 (D17) and 76 (D76) and glutamic acid 126 (E126) within the transmembrane region of Tim17. **b**, Import of radiolabelled $F_1\beta$ (matrix), $b_2(167)_\Delta$-DHFR (matrix) and $b_2(220)$-DHFR (inner membrane-sorted) precursors into isolated mitochondria; subsequently, they were treated with proteinase K and subjected to SDS–PAGE and autoradiography. i*, intermediate. **c**, MTX preincubated radiolabelled $b_2(220)$-DHFR was imported into isolated wild-type Tim17 or mutant mitochondria followed by BN–PAGE and autoradiography. TOM-$b_2(220)$-DHFR, MPP and IMP processed $b_2(220)$-DHFR was stabilized at the TOM complex.

subunits remains similar to the first purification of the translocase, we observe an intense enrichment of Jac1–GFP and of TOM complex subunits in the second purification. This indicates that higher-order Tim17–Tim23 heterodimers are not stably associated, neither in the presence nor absence of engaged precursor. Together, these results and the structural modelling data (Extended Data Fig. 4) support the view that Tim17 and Tim23 do not form stable channel-like structures by direct association of their lateral cavities.

## Negative charges of Tim17 are essential

A sequence alignment of Tim17 family proteins indicated the highest conservation of negative charges within the transmembrane domains of Tim17 itself, located on the intermembrane space side of the transmembrane domains within the lateral cavity (Fig. 3a and Extended Data Fig. 5a,b). We generated single-point mutants of the three conserved negative charged residues within the transmembrane domains of Tim17 in yeast. Arginine (charge reversal) or phenylalanine (large residue) mutants exhibited a very strong growth phenotype (specifically for D17 and D76; for example, no growth on non-fermentable glycerol medium (YPG))[27] (Extended Data Fig. 5c and Extended Data Table 1). By contrast, replacement with small residues such as alanine did not cause obvious growth defects. Therefore, we selected the well-tolerated alanine mutants to perform functional assays. Isolated Tim17[D17A] and Tim17[D76A] mitochondria had protein levels and a TIM23[core] complex assembly comparable with wild-type Tim17 (Extended Data Fig. 6a,b). Moreover, the membrane potential supported assembly of the metabolite carrier protein Dic1 comparable with wild-type protein (Extended Data Fig. 6c,d). To assess whether the negatively charged residues of Tim17 are crucial for mitochondrial biogenesis, we performed import of presequence-containing proteins. Import of matrix-destined and inner membrane-sorted precursor proteins was slightly affected in the D76A mutants and especially pronounced in the D17A mutant mitochondria (Fig. 3b). The passenger domain of the $b_2(220)$-DHFR is imported into the mitochondrial intermembrane space (the precursor is processed by MPP, sorted into the inner membrane and subsequently cleaved a second time by the mitochondrial inner membrane protease). Upon import of $b_2(220)$-DHFR in the presence of the DHFR ligand MTX, the fully processed precursor accumulates at the TOM complex[14]. This assay confirms that presequence protein import is affected in the Tim17[D76A] mutant and

even more pronounced in the Tim17[D17A] mutant (Fig. 3c). It is tempting to speculate that Tim17 proteins of higher eukaryotes contain two directly adjacent negatively charged residues at positions equivalent to residues 16 and 17 in yeast owing to the crucial role of this negatively charged site for the import of presequence proteins (Extended Data Fig. 5a). To analyse whether the negative charged residues of Tim17 contact translocating precursor proteins, we generated the cysteine mutants Tim17[D76C] and Tim17[E126C] (in agreement with Extended Data Fig. 5c and Extended Data Table 1, the Tim17[D17C] yeast strain displayed strong growth defects and was, therefore, not further analysed) (Extended Data Fig. 6e). Cysteine-specific crosslinking of arrested matrix-destined precursor proteins confirmed that they are translocated in the immediate vicinity of the conserved negatively charged residues of Tim17 (Extended Data Fig. 6f). All three Tim17 D17, D76 and E126 double alanine mutants did not grow on non-fermentable medium (Extended Data Table 1). A Tim17[D17A_D76A_E126A] triple mutant was not viable, indicating that the conserved negative charges within the lateral cavity of Tim17 together are essential for presequence protein translocation across the inner membrane (Extended Data Table 1). By contrast, Tim23 mutants lacking the negative charges within the transmembrane and adjacent segments grow similar to wild type (Extended Data Fig. 6g). Moreover, the Tim17[D17A_D76A_E126A] triple mutant could be rescued neither by introducing analogous charges into the lateral cavity of Tim23 nor by introducing equivalent negative charges into Tim17 placed approximately two or three turns closer toward the middle of the bilayer (Extended Data Fig. 6h,i). Tim17 mutants with additional negative charges within the lateral cavity were also not viable (Extended Data Fig. 6i).

## Tim17 translocation initiation site

To analyse severe Tim17 mutants, we placed the chromosomal copy of *TIM17* under the control of the galactose promotor (pGAL-*TIM17*) and expressed the *TIM17* alleles of interest from an essential plasmid on non-fermentable glycerol medium (YPG) (Extended Data Fig. 7a,b). Growth of these strains on YPG agar plates was strongly affected for the Tim17[D76A_E126A] mutant and even more pronounced for the Tim17[D17A_D76A] and Tim17[D17A_E126A] mutants (Fig. 4a). The initial growth of the Tim17[D17A_D76A_E126A] triple mutant was comparable with the empty vector control, confirming that the conserved negative transmembrane charges are essential for viability. We selected the Tim17[D17A_D76A] mutant

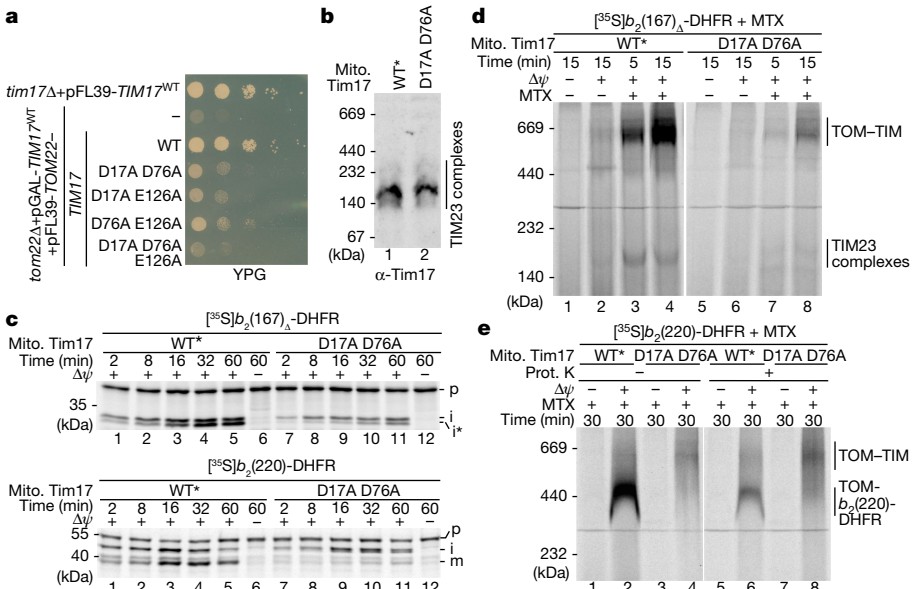

**Fig. 4 | Tim17-dependent presequence translocation is initiated at the negative charged patch. a**, Growth analysis of yeast strains expressing wild-type or *tim17* variants with the indicated double or triple negative charge mutation on non-fermentable glycerol (YPG) medium at 19 °C. **b**, Mitochondrial protein complexes were analysed by BN–PAGE as described in Fig. 1f. WT\*, mitochondria isolated from *tom22*Δ, pGal-*TIM17*[WT] + pFL39-*TOM22* yeast cells with levels of Tim17 comparable with Tim17[D17A_D76A] mutant mitochondria.

**c**, Protein import into wild-type and Tim17[D17A_D76A] mitochondria was analysed as described for Fig. 3c. **d**, Radiolabelled $b_2(167)_\Delta$-DHFR was imported and accumulated in import sites of isolated mitochondria in the presence of MTX as indicated. Samples were analysed by BN–PAGE and autoradiography. TOM–TIM, TOM–TIM23 preprotein stabilized supercomplex. **e**, Accumulation of radiolabelled $b_2(220)$-DHFR in TOM–TIM23 import sites of isolated mitochondria as in Fig. 3c, followed by treatment with proteinase (Prot.) K.

for further analysis because the previous results indicated that those two residues are the most critical residues and because the Tim17[D17A_D76A] protein was more stable compared with the other mutant variants. To isolate mitochondria at permissive conditions, a time point was selected at which the growth of the Tim17[WT], the Tim17[D17A_D76A] mutant and the empty vector control strains was comparable with each other (Extended Data Fig. 7c). We isolated mitochondria of the Tim17[D17A_D76A] strain and of Tim17[WT*] with similar protein levels of Tim17, further TIM23 subunits and TIM23 complexes (Fig. 4b and Extended Data Fig. 7d–f). The membrane potential of the Tim17[D17A_D76A] mitochondria and their ability to assemble the TIM22-dependent carrier protein Aac2 were not affected compared with wild-type mitochondria (Extended Data Fig. 7g,h). Import of presequence-containing precursor proteins in Tim17[D17A_D76A] mitochondria was strongly diminished compared with wild-type mitochondria, no matter if matrix-destined or inner membrane-sorted precursors were analysed (Fig. 4c and Extended Data Fig. 7i). These results were confirmed by analysing the import and assembly of presequence-containing precursor protein–DHFR fusion constructs in the presence of MTX. The formation of the TOM–TIM23 supercomplex by arrest of $b_2(167)_\Delta$-DHFR[18] and the formation of the TOM-$b_2(220)$-DHFR were strongly reduced in the Tim17[D17A_D76A] mitochondria (Fig. 4d,e). These results unambiguously demonstrate that the negative charges within the lateral transmembrane cavity of Tim17 are crucial for the translocation of the precursor protein across the inner membrane.

## Translocation path for preproteins

Our $b_2$-DHFR precursor crosslinking assays indicated that the arrested precursor proteins are specifically crosslinked to Tim17 transmembrane domain (Fig. 1c and Extended Data Fig. 1). However, an exact mapping of the position of the crosslinks to Tim17 was not possible using this in organello assay. To perform a position-specific crosslinking of mitochondrial presequence precursor proteins with Tim17

transmembrane portions, we inserted single cysteines along the lateral cavities in Tim17[SCF] and Tim23[CF] backgrounds (Fig. 5a and Extended Data Fig. 8a). Arrest of matrix- and inner membrane-sorted precursor proteins followed by cysteine-specific crosslinking revealed a similar crosslinking pattern with Tim17[K36C] and Tim17[N64C] (Fig. 5b,c and Extended Data Fig. 8b), whereas no comparable crosslinks were observed with cysteines located within the lateral cavity of Tim23 (Extended Data Fig. 8c–e). To test whether the presequence precursor proteins can also be crosslinked within the lateral Tim17 cavity in the absence of the inner membrane potential, we incubated the precursor proteins in the presence and absence of a membrane potential followed by cysteine-specific crosslinking. The membrane potential-dependent crosslink formation (Extended Data Fig. 8f) indicated that presequence-containing precursor proteins cross the inner membrane in a membrane potential-dependent manner at the lateral cavity of the Tim17 membrane domain. This analysis was confirmed by a cysteine-bispecific crosslinking approach after arresting the matrix- and inner membrane-destined precursors with single-cysteine residues in the sorting signal (used in Fig. 1 and Extended Data Fig. 1) in the mitochondria with single cysteines along the lateral cavities of Tim17[SCF] or Tim23[CF] (used in Fig. 5b,c and Extended Data Fig. 8b–f). This analysis demonstrated a gradual change of the site-specific crosslinking efficiencies, which depend on the localization of the specific cysteines within the precursor protein and within the Tim17 lateral cavity. Although arrested $b_2$-DHFR precursor protein constructs with cysteines at positions 47 and 49 mainly formed crosslinks with Tim17[K36C] and Tim17[N64C], precursors with cysteines at position 51 or 54 were additionally crosslinked to Tim17[D76C] (Fig. 5d and Extended Data Fig. 8g,h). This analysis confirms that arrested precursor proteins are associated along the lateral cavity of Tim17 from D76 (one of the conserved negative charges of Tim17 located on the intermembrane space side within the lateral cavity in TM2) over N64 (hydrophilic residue within the lateral cavity crucial for matrix translocation (Fig. 1e–g and Extended Data Fig. 2) in the middle of Tim17–TM2) to K36 (a residue at

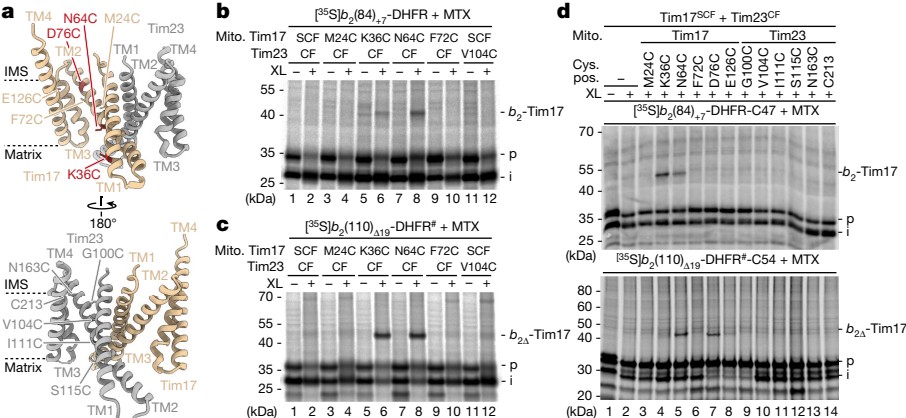

**Fig. 5 | Arrested precursor proteins are associated throughout the lateral cavity of Tim17. a**, ColabFold protein complex model of *S. cerevisiae* Tim17 (tan) and Tim23 (grey) heterodimer showing the location of cysteine mutants (tan/dark red/grey) within the transmembrane regions of Tim17 (top) or Tim23 (bottom). **b,c**, Radiolabelled MTX-stabilized $b_2(84)_{+7}$-DHFR (**b**) and $b_2(110)_{\Delta 19}$-DHFR (**c**) constructs were accumulated in mitochondria with cysteine residues in Tim17 (SCF) or Tim23 (CF) at TOM–TIM23 import sites. Subsequently, samples were chemically crosslinked with MBS (XL) and analysed by SDS–PAGE and autoradiography. $b_{2\Delta}$-Tim17, $b_2(110)_{\Delta 19}$-DHFR-Tim17 crosslinking product. **d**, $b_2$-DHFR constructs containing a cysteine at position 47 (C47; $b_2(84)_{+7}$-DHFR) or 54 (C54; $b_2(110)_{\Delta 19}$-DHFR) were analysed as described in **c** but chemically crosslinked with BMOE (XL).

the matrix side of the lateral cavity located in Tim17–TM1). Together, our results support the view that the translocation path across the inner membrane through the Tim17 cavity starts at the conserved negatively charged patch close to the intermembrane space side and continues through the cavity along the extended TM1 of Tim17 to reach the mitochondrial matrix.

## Mitochondrial presequence translocation

Based on our results, we propose that mitochondrial presequences are translocated across the inner membrane at the Tim17 bilayer interface. However, the prevailing doctrine based on numerous publications from different groups was the assumption that the Tim17 protein family members Tim23 and Tim22 are the core subunits of the presequence and carrier translocases, which form hydrophilic channels for precursor protein translocation across and/or for membrane insertion into the inner mitochondrial membrane[8–12,35–37]. To support this assumption, over the years several sophisticated methods were used, which accumulated data in support of the Tim23 channel model (for example, refs. 11,38). It cannot be disputed that the major Tim17 family proteins Tim17, Tim22 and Tim23 all form assemblies with channel-like properties. Martinez-Caballero et al.[39] performed patch clamping of TIM23 from reconstituted inner membranes and observed two equal-sized 'pores', which show cooperative gating behaviour. Depletion of Tim17 induced a collapse of the twin pores into a single pore[39]. Therefore, we speculate that the narrow lateral transmembrane cavities of Tim17 and Tim23 (presequence translocase) and Tim22 (carrier translocase) all show channel-like properties in electrophysiological experiments, which led to the current models of mitochondrial inner membrane translocation and insertion through hydrophilic channels[1,2,5–7]. To reflect on the translocation mechanism across the inner membrane, we consider the properties of mitochondrial presequences. (1) Presequences can form an amphiphilic helix and perturb the phospholipid bilayer[40]. (2) Amphiphilicity is essential for mitochondrial presequence function[41]. (3) Interaction of the precursor (that is, presequence) with a lipid bilayer may represent a key aspect of protein import into mitochondria[42]. These properties are not required for translocation through an aqueous channel. The amphiphilic character of mitochondrial presequences rather directly matches a Tim17-dependent translocation mechanism at the bilayer interface. As a secondary consequence, the properties of the presequence receptor Tom20 (hydrophobic groove lined with a negatively charged residue[43]) and of the protein-conducting

β-barrel channel Tom40 (hydrophobic and negatively charged patches within the aqueous channel[44]) of the TOM complex evolved to recognize and translocate the positively charged amphiphilic presequences across TOM. The intermembrane space domains of the TIM23 subunits Tim50, Tim23 and Tim21 mediate the recognition and transfer of the presequences to the inner membrane[1,2,5,6]. In addition, two conserved negative charges in the N terminus of Tim17 immediately adjacent to the inner membrane (N-terminal of TM1) were reported to be crucial for the import of presequence proteins[45]. The non-essential TIM23 subunit Mgr2 (implicated in regulation of lateral sorting) is predicted by co-evolution analysis and structural modelling to shield the lateral cavity of Tim17 and to form a protein enclosed environment within the bilayer reminiscent of a channel-like structure but with a hydrophobic interior surface[14,26,46] (Fig. 6a and Extended Data Fig. 9a,b). Modelling of the Tim17–Tim23–Mgr2 heterotrimer including a presequence with ColabFold, generated an assembly where the presequence is specifically associated at the Tim17 lateral cavity[47] (Extended Data Fig. 9c). However, it remains to be determined if Mgr2 really associates with the lateral cavity of Tim17 to shield the translocating precursor protein from the bilayer. Enhanced dissociation from the Tim17–Tim23 heterodimer in the absence of precursor protein could help to preserve the permeability barrier for the maintenance of the mitochondrial membrane potential. Because mitochondrial protein translocation also works in the absence of the non-essential lateral gate keeper subunit Mgr2 (ref. 48), we propose the following basic model for the translocation of mitochondrial presequences across the inner membrane (Fig. 6b). The acidic patch of the lateral transmembrane cavity of Tim17 is located close to the intermembrane space side, positioned to attract the positive charge of the N terminus of the precursor protein. Therefore, we name this acidic patch the Tim17 translocation initiation site (TIS). Concomitantly, the hydrophobic residues of the amphiphilic presequence engage with the outer leaflet of the lipid bilayer, which stabilizes the α-helical conformation of the presequence and reinforces the interaction between the presequence and the TIS. On the hydrophilic side, mitochondrial presequences are rich in further positive charged residues, which are frequently three to four positions apart such that they occur at roughly every turn and face the same side of the helix[49]. These subsequent positively charged residues on the hydrophilic side of the α-helical presequence enable insertion of the hydrophobic face into the lipid bilayer in steps of single turns. The negatively charged phospholipid head groups on the intermembrane space side and on the matrix face likely support the translocation. Presequences have

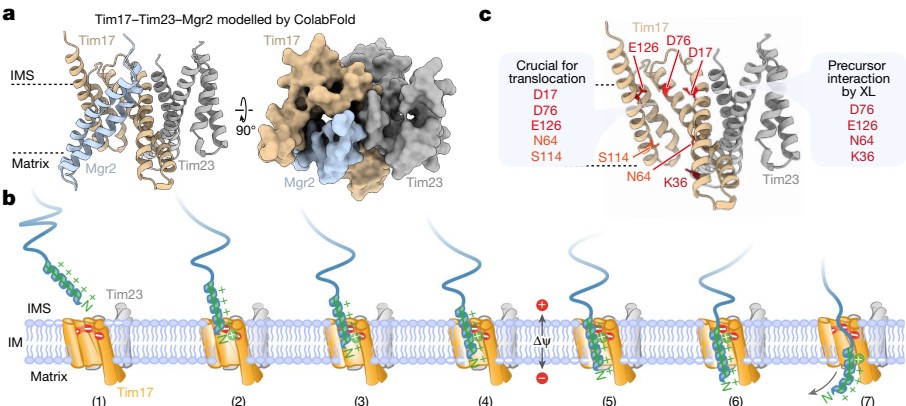

**Fig. 6 | Model for mitochondrial presequence translocation across the inner mitochondrial membrane. a**, Structural protein complex model of the interaction between Tim17 (tan), Mgr2 (blue) and Tim23 (grey) generated by ColabFold, shown as cartoon (left) and surface (right) representations. **b**, The Tim17 TIS composed of the acidic patch of the lateral transmembrane cavity of Tim17 is positioned close to the intermembrane space side and attracts the positive charge of the N terminus of the precursor protein (1). The hydrophobic residues of the amphiphilic presequence engage with the outer leaflet of the lipid bilayer, which stabilizes the α-helical conformation of the presequence and reinforces the interaction between the presequence and the TIS (2). Successive positively charged residues on the hydrophilic side of the α-helical presequence enable insertion of the hydrophobic face into the lipid bilayer in steps of single turns (3 and 4). The negatively charged phospholipid head groups on the matrix face of the inner membrane attract the positively charged amino terminus (5). The membrane potential (Δψ) and following positive residues support the emergence of the presequence on the matrix side (6) and engagement with the mitochondrial presequence translocase-associated motor, which aids the remaining steps of import into the matrix (7). **c**, Summary of the findings of the current study depicted onto the ColabFold structural model of the interaction between Tim17 (tan) and Tim23 (grey). Red, residues of the Tim17 TIS; orange, hydrophilic residues; dark red, residues crosslinked to arrested precursor protein intermediates.

a predominant length between 15 and 40 residues and can span the thinned inner membrane at the Tim17 cavity, which was similarly observed for other translocases and insertases[3,27,50]. Precursor proteins with shorter presequences, like the ATP synthase subunit Atp17/Suf, are compatible with the same mechanism, as they often contain presequence-like features within their mature region adjacent to the presequence. The mitochondrial membrane potential (Δψ) is essential for presequence protein translocation[51,52]. We propose that Δψ is crucial to support the initial extraction of the mitochondrial presequences from the inner membrane such that the essential presequence translocase-associated import motor (PAM) subunit mtHsp70 can bind to the first hydrophobic part of the presequence after it emerges into the matrix. This model of mitochondrial protein translocation across the inner membrane is based on functional defects of *tim17* point mutants and site-specific crosslinking along the lateral cavity of Tim17 from the intermembrane space side to the mitochondrial matrix (Fig. 6c). So far, we discussed how the presequence is translocated across the inner membrane and envision three major cases of how the mature part of the precursor is translocated. (1) The mature part of the precursor has presequence-like features, and translocation can proceed as described. (2) The subsequent mature part of the precursor has chemical characteristics of side chains, which are not attracted by the Tim17 lateral cavity. In case the mature part has adopted an α-helical conformation, we envision that the import motor driving force will cause an unfolding of the precursor secondary structure on the intermembrane space side[53,54]. The translocation of the extended chain along the narrow lateral cavity of Tim17 is supported by hydrophilic residues (Fig. 1e–g), and association with Mgr2 might reduce the contact to the bilayer as much as possible. (3) The translocation at the Tim17 bilayer interface enables the lateral release of a subsequent transmembrane domain/stop-transfer signal of inner membrane-sorted proteins likely by dissociation of Mgr2, in agreement with the report that Mgr2 modulates the threshold hydrophobicity for membrane insertion[55].

Together, we believe that our detailed site-specific interaction studies that map the precursors to the Tim17 lateral cavity and vice versa, the back-to-back conformation of the Tim17–Tim23 heterodimer as a functional unit for precursor translocation, the import defects using *tim17* mutants and the additional evidences presented above are sufficient to make the following three major conclusions. (1) Tim17 (and not Tim23) is the major subunit of the presequence translocase directly involved in translocation of presequence proteins across the inner membrane. (2) Mitochondrial presequence proteins can be imported across the inner membrane at the Tim17 bilayer interface. The negatively charged patch on the intermembrane space side of the lateral transmembrane cavity of Tim17 acts as a TIS to import the presequences along the cavity across the inner membrane. (3) We propose that the essential Tim22 core subunit of the carrier translocase, required for the biogenesis of mitochondrial metabolite carrier proteins, operates by using a similar membrane insertion mechanism at the Tim22 bilayer interface.

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

# Methods

## Yeast strains, plasmids and primers

The *S. cerevisiae* strains, plasmids and primers used in this study are listed in Supplementary Tables 1–3, respectively. The Tim17 shuffle strain (YPH499 *tim17Δ* + YEp352-*TIM17*) was used to generate yeast strains expressing wild-type Tim17 or mutant proteins using the plasmid shuffling approach[21]. Transformation of the *TRP1* selectable pFL39-Tim17 (containing native promoter and terminator regions) or related plasmids encoding the relevant mutant version followed by selection on 5-fluoroorotic acid (5-FOA) plates (9.33 mM 5-FOA, 0.067% (wt/vol) yeast nitrogen base without amino acids (Difco), 0.077% (wt/vol) complete supplement mixture (CSM) amino acids without uracil (MP Biomedicals), 0.005% (wt/vol) uracil (Sigma), 2–3% (wt/vol) glucose and 3% (wt/vol) agar (Becton Dickinson) resulted in removal of the wid-type copy and exchange with the mutant. Mutants were considered lethal if no colonies were recovered from 5-FOA plates. The Tim17 galactose regulatable strain was generated by homologous recombination of a PCR cassette containing the GAL promoter and antibiotic resistance for kanamycin (*KanMX6*) flanked by regions upstream and downstream of the *tim17* promoter. Chromosomal integration of the GAL promoter (pGAL) upstream of *tim17* in *tom22Δ* + YEp352-*TOM22* (refs. 56,57) led to the generation of *tom22Δ*, pGAL-Tim17 + YEp352-*TOM22*. To study severe Tim17 mutants in this strain, *tim17* wild-type or mutant with native promoter and terminator regions was amplified by PCR from pFL39-*TIM17* and subsequently introduced into pFL39-*TOM22* (*TRP1* selectable) using the restriction enzyme EcoRI (NEB). This plasmid was transformed into *tom22Δ*, pGal-*TIM17* + YEp352-*TOM22* using the plasmid shuffling approach described above with the exchange of glucose with 2% (wt/vol) galactose during selection. To isolate the TIM23 complex, a Protein A tag was introduced downstream of *tim21* in *tom22Δ*, pGal-*TIM17* + pFL39-*TOM22-TIM17 WT* or *D17A_D76A* mutant by homologous recombination as described[57]. The Tim23 shuffle and Tim17–Tim23 double-shuffle strain were generated by deletion of *TIM23* (and *TIM17*) by homologous recombination complemented by wild-type *TIM23* (and wild-type *TIM17*) on URA3 selectable plasmids (YPH499 *tim17Δ tim23Δ* + YEp352-*TIM17*, YEp352-*TIM23*)[21,57]. To study Tim17 and Tim23 cysteine mutants in a Tim17 SCF and Tim23 CF background, *tim23^CF* with native promoter and terminator regions was amplified by PCR from pFL39-*TIM23^CF* and inserted into pFL39-*TIM17^SCF* (*TRP1* selectable) using EcoRI (NEB). Tim17 and Tim23 cysteine mutants were subsequently made in this vector and introduced into the Tim17–Tim23 double-shuffle strain by the plasmid shuffling described above. To create Tim17^2xStrep- and ^HisSUMO*Tim23-expressing strains, a glycine–alanine–glycine linker (GGGGAGGGG) and 2xStrep tag were placed downstream of the *TIM17* open reading frame (ORF). This region along with native promoter and terminator was amplified by PCR and ligated into pFL39-*TOM22* using EcoRI (NEB) to generate pFL39-*TOM22-TIM17^Linker-2xStrep*. Tim17^Linker-2xStrep is referred to as Tim17^2xStrep throughout the study. ^HisSUMO*Tim23 with native promoter and terminator was amplified from R737 (plasmid; a gift from the Rehling group) and introduced into pFL39-*TIM17^2xStrep* to generate either pFL39-*TOM22-^HisSumo*TIM23* or pFL39-^HisSumo*TIM23-TIM17^Linker-2xStrep*.

Yeast cells were grown in YPG (1% (wt/vol) yeast extract, 2% (wt/vol) bacto peptone, 3% (wt/vol) glycerol, pH 5.0), YPGal (1% (wt/vol) yeast extract, 2% (wt/vol) bacto peptone, 2% (wt/vol) galactose) or YPD (1% (wt/vol) yeast extract, 2% (wt/vol) bacto peptone, 2% (wt/vol) glucose) at 19–23 °C and 130 rpm. For growth analysis, yeast cells were grown in YPG, YPGal or YPD overnight at 23 °C, and a volume corresponding to an $OD_{600}$ of one was collected and resuspended in sterile water. The cells were plated at a fivefold serial dilution on YPG or YPD plates and incubated at 19–36 °C. For liquid culture growth analysis of Tim17-inducible strains, yeast cells were grown in YPGal overnight at 23 °C, and a volume corresponding to an $OD_{600}$ of 0.5 was taken. Cells were washed twice in YPG and inoculated at an $OD_{600}$ of 0.1. Plate reader growth assays were performed on a ClarioStar microplate reader (BMG LabTech), and data were collected for up to 90 h.

Amino acid substitutions of Tim17 or Tim23 were introduced by site-directed mutagenesis using the centromeric plasmid pFL39 containing the native promoter, terminator and wild-type open reading frame of *S. cerevisiae TIM17* or *TIM23* (Supplementary Table 2) as a template for PCR amplification with primers encoding the desired mutation(s) (Supplementary Table 3) using KOD high-fidelity polymerase (Novagen). The plasmid template was digested (37 °C, 3 h) with DpnI (NEB), and the PCR products were subsequently transformed into competent XL-1 Blue *Escherichia coli* cells. Single colonies were selected, and plasmids were prepared using the QIAprep Spin Miniprep Kit (Qiagen) and verified by sequencing.

For Gibson cloning, individual DNA fragments were generated by PCR using KOD Hot Start DNA Polymerase. The resulting PCR products were incubated with DpnI (NEB) at 37 °C for 1–3 h to remove template DNA. The PCR products were purified with the QIAquick PCR purification kit (Qiagen) and subsequently ligated using Gibson Assembly Master Mix (NEB) according to the manufacturer's instructions.

Introduction of individual cysteines in $b_2(84)$-DHFR and $b_2(110)_Δ$-DHFR was done by single-primer site-directed mutagenesis. Plasmids pGEM4Z-$b_2(84)$-DHFR and pGEM4Z-$b_2(110)_Δ$-DHFR were used as templates for single-primer PCR reactions with Q5 polymerase master mix (NEB) and primers encoding the mutations (Supplementary Table 3). After the PCR reaction, matching reverse and forward PCRs were mixed, denatured and slowly annealed to receive double-stranded products containing the desired mutation. After 2 h of DpnI (NEB) digestion at 37 °C, the reactions were transformed into competent MachT1 *E. coli* cells. Single colonies were picked and inoculated in ampicillin- or carbenicillin-containing LB medium, and the plasmids were isolated using the QIAprep Spin Miniprep Kit (Qiagen). Positive plasmids were identified by sequencing.

## Isolation of mitochondria

Yeast cells were grown in YPG at 23 °C and 130 rpm until an early logarithmic growth phase. Mitochondria were isolated by differential centrifugation[58]. Cells were harvested by centrifugation (2,500*g*, 10 min), washed in water and subsequently incubated in 2 ml per gram of cells prewarmed DTT softening buffer (100 mM Tris-$H_2SO_4$, pH 9.4, 10 mM DTT) for 20 min at growth temperature. Cells were washed and incubated in 6.5 ml per gram of cells zymolase buffer (16 mM $K_2HPO_4$, 3.96 mM $KH_2PO_4$, pH 7.4, 1.2 M sorbitol (Roth)) containing 3.5 mg zymolase per gram of yeast cells at 23 °C for 30–40 min. Spheroplasts were washed once in zymolase buffer (without zymolase) and cells were mechanically opened on ice with a Teflon glass homogenizer in cold homogenization buffer (0.6 M sorbitol, 10 mM Tris-HCl, pH 7.4, 1 mM ethylenediaminetetraacetic acid (EDTA), 0.4% (wt/vol) bovine serum albumin (Sigma), 1 mM phenylmethylsulfonyl fluoride (PMSF; Sigma). Cell debris was removed from the homogenate by consecutive centrifugation (150*g*, 2 min; 250*g*, 2 min and 1,000*g*, 4 min; 1,800*g*, 5 min at 4 °C), and mitochondria were collected from the supernatant (16,000*g*, 15 min, 4 °C). Mitochondrial pellets were washed in cold SEM buffer (10 mM MOPS/KOH, pH 7.2, 250 mM sucrose, 1 mM EDTA) and clarified (1,800*g*, 5 min at 4 °C). Mitochondria were subsequently reisolated (16,000*g*, 15 min, 4 °C), resuspended in SEM buffer to a protein concentration of 10 mg ml$^{-1}$, snap frozen as aliquots in liquid nitrogen and stored at −80 °C until use.

## Protein expression and purification

Recombinantly expressed Jac1–sfGFP was purified as described by Gomkale et al.[20] with modifications. In brief, plasmids (a gift from the Rehling group) containing the gene for overexpression of 14His–SUMO–Jac1–sfGFP were transformed into competent Rosetta (BL21 DE3) cells and cultured overnight (37 °C, 180 rpm) before dilution to an $OD_{600}$ of 0.1. Expression was induced when cells reached an

OD$_{600}$ of 0.6 by addition of 1 mM IPTG at 37 °C and 180 rpm. After 4 h, cells were collected by centrifugation (2,500$g$, 15 min). Purification was performed on an ÄKTA pure system (Cytiva). Cells were resuspended in buffer A (40 mM Tris-Cl, pH 7.4, 500 mM NaCl, 10 mM imidazole), lysed by sonication and centrifuged (16,000$g$, 20 min), and the filtered supernatant was applied to a HisTrap HP column (Cytiva) pre-equilibrated with buffer A. Elution was performed using a linear gradient of buffer B (40 mM Tris-Cl, pH 7.4, 500 mM NaCl, 500 mM imidazole). Protein-containing fractions were analysed by SDS–PAGE and Coomassie brilliant blue staining. Relevant fractions were pooled and dialysed overnight at 4 °C against buffer A2 (20 mM Tris-HCl, pH 7.4, 100 mM NaCl). Removal of the 14His-SUMO* tag utilized for purification was achieved by overnight incubation with His-tagged SUMO protease (20 U mg$^{-1}$ protein in the presence of 1 mM DTT) at 4 °C. To remove uncleaved 14His–SUMO–Jac1–sfGFP, His-tagged SUMO protease and 14His–SUMO* tag, the mixture was applied to the HisTrap HP column again using buffer A2 for equilibration and buffer B for regeneration of the column. Flow-through fractions were collected, and the protein content was analysed by SDS–PAGE and staining with Coomassie brilliant blue. Protein concentration was determined, and purified protein was snap frozen in aliquots and stored at −20 °C.

## Mitochondrial protein import and formation of the TOM–TIM23 supercomplex

Radiolabelled mitochondrial precursor proteins were generated using a TNT SP6 Coupled Reticulocyte Lysate System (Promega) with plasmid DNA, according to the manufacturer's instructions. In vitro translation was performed with plasmids carrying the gene of interest downstream of the SP6 promoter as a template in the presence of [$^{35}$S]methionine. For synthesis of radiolabelled Cox5a–sfGFP and Dic1, the open reading frame fused to the SP6 promoter sequence was amplified by PCR. mRNA transcripts were produced using the mMESSAGE mMACHINE transcription kit (Promega) and purified with the MEGAclear transcription cleanup kit (Thermo Fisher). mRNA transcripts were used for in vitro translation in the presence of [$^{35}$S]methionine using the TNT SP6 Coupled Reticulocyte Lysate System or the Flexi Rabbit Reticulocyte Lysate System (Promega).

Isolated mitochondria were incubated with radiolabelled or recombinant precursor proteins in import buffer (3% (wt/vol) fatty acid free bovine serum albumin, 10 mM MOPS/KOH, pH 7.2, 250 mM sucrose, 80 mM KCl, 5 mM MgCl$_2$, 5 mM methionine, 2 mM KH$_2$PO$_4$) containing 4 mM ATP (Roche), 4 mM NADH (Roth), 2 mM creatine phosphate (Roche) and 2 mM creatine kinase (Roche). To deplete the membrane potential (−Δ$\psi$), AVO mix (8 μM antimycin A (Sigma), 1 μM valinomycin (Sigma) and 20 μM oligomycin (Sigma)) was added, whereas NADH was excluded; 6–10% (vol/vol) of rabbit reticulocyte lysate was added, and import reactions were incubated at 25 °C and 300 rpm for the indicated time periods. Import was stopped by adding AVO and placing samples on ice. Where indicated, samples were treated with proteinase K (50 μg ml$^{-1}$) for 10 min on ice followed by the addition of 2 mM PMSF for 5–10 min on ice to inhibit protease activity. Translocation arrest of precursor proteins was achieved by folding and stabilization of the DHFR domain with 5 μM MTX[18,59]. Mitochondria were isolated (20,000$g$, 5 min), washed in cold SEM buffer and analysed by either BN or denaturing gel electrophoresis.

The formation of an import intermediate by stable arrest at the TOM–TIM23 supercomplex was performed as previously described[14,20,21]. Radiolabelled Cox5a(1–130)–sfGFP and DHFR-containing precursor protein fusions $b_2$(167)$_{\Delta19}$-DHFR, $b_2$(220)-DHFR, $b_2$(84)$_{+7}$-DHFR and $b_2$(110)$_{\Delta19}$-DHFR_C86S ($b_2$(110)$_{\Delta19}$-DHFR$^{\#}$) or cysteine-containing variants (C47 to C54) were initially incubated for 5–10 min at 25 °C in BSA-containing import buffer (as described above) supplemented when required with 5 μM MTX to permit folding of the DHFR domain. For certain experiments, AVO mix was additionally added to the reaction. For import of recombinantly expressed Jac1–sfGFP, purified protein

was diluted to a concentration of 40 μM in 20 mM Tris-Cl, pH 7.4, and 100 mM NaCl and added to import reactions to a final concentration of 2.5 μM. Subsequently, mitochondria were added, and the import reactions were incubated for 30 min at 25 °C (300 rpm). Protein import reactions were stopped by addition of AVO mix, and mitochondria were isolated by centrifugation (20,000$g$, 5 min, 4 °C). If mitochondria were subsequently used for crosslinking or affinity purification, the individual reactions were layered over an S$_{500}$EM (10 mM MOPS/KOH, pH 7.2, 500 mM sucrose, 1 mM EDTA) cushion (equal to or double the total import reaction volume) followed by centrifugation (20,000$g$, 15 min, 4 °C). The resulting mitochondrial pellets were additionally washed with SEM buffer to remove residual radiolabelled protein lysate (20,000$g$, 5 min, 4 °C).

Heat shock of Tim17 temperature-sensitive mutants was performed as previously described[21]. Wild-type Tim17, *17-4*, *17-5* and Tim17$^{2xStrep}$-$^{HisSUMO*}$Tim23 mitochondria were incubated in import buffer (lacking NADH, ATP, CP and CK) at 37 °C for 10 min. Heat-shocked mitochondria were directly added to complete import buffer, and import was performed at 25 °C and 300 rpm.

## In vitro oxidation of mitochondria

Mitochondria were incubated in 100 μl of SEM buffer supplemented with 2 mM PMSF and 1× protease inhibitor (EDTA free; Roche) and treated with 0.2 mM 4-DPS for 10 min on ice. Oxidation reactions were stopped by the addition of 50 mM iodoacetamide and 10 mM EDTA and incubated on ice for 10 min. Mitochondria were isolated by centrifugation (20,000$g$, 10 min, 4 °C), resuspended in reducing (containing 50 mM DTT) or non-reducing Laemmli buffer and analysed by SDS–PAGE followed by western blotting.

## In organello crosslinking

Following formation of an arrested import intermediate at TOM–TIM23 import sites, washed mitochondrial pellets were resuspended in fresh SEM buffer, aliquoted and supplemented with DMSO (no crosslinker control), DMSO-solubilized MBS (Thermo Fisher; 1 mM final concentration) or BMOE (Thermo Fisher; 1 mM final concentration) for 30 min on ice. The reactions were quenched by the addition of quencher (SEM buffer supplemented with 100 mM Tris-HCl, pH 7.4, and 100 mM iodoacetamide) at a 1:1 ratio for MBS crosslinking or iodoacetamide (resuspended in distilled water) to a final concentration of 50 mM for BMOE crosslinking approaches. Next, the quenched samples were centrifuged (20,000$g$, 15 min, 4 °C) and washed again with fresh SEM buffer (20,000$g$, 5 min, 4 °C). The washed mitochondrial protein pellets were resuspended in 2× Laemmli buffer supplemented with 2 mM PMSF at a ratio of 2 μg protein per 1 μl and denatured at 65 °C (1,400 rpm, 15 min). The resulting samples were loaded onto 10% Tris-Tricine SDS–PAGE gels (total amount per lane: 70–100 μg mitochondrial protein) and run at 120 V and 30 mA (per gel) for 14.5–16 h. Proteins were semi-dry blotted onto a polyvinylidene difluoride (PVDF) membrane, or protein-containing gels were dried. Membranes and gels were exposed to Phosphoimager screens and analysed by autoradiography.

## Affinity purification of protein complexes or crosslinking products

For Protein A purification of the TIM23 complex from Tim17 WT*/Tim21$^{ProtA}$ and D17A_D76A/Tim21$^{ProtA}$, mitochondria were solubilized in digitonin solubilization buffer (20 mM Tris-Cl, pH 7.4, 50 mM NaCl, 0.1 mM EDTA, 10% (vol/vol) glycerol, 1× protease inhibitor cocktail (cOmplete Tablets EDTA free plus Easy pack; Roche) and 2 mM PMSF) containing 1% (wt/vol) digitonin end over end at 4 °C for 30 min. Clarified mitochondrial lysates were combined with equilibrated IgG Sepharose and incubated end over end at 4 °C for 2 h. Bound protein complexes were washed (10 times column volume) and subsequently incubated with 150 U tobacco etch virus protease (Invitrogen) in solubilization buffer containing 0.3% (wt/vol) digitonin at 4 °C for 16 h and

1,000 rpm. Tobacco etch virus protease was removed from the elution fraction by incubation with equilibrated Ni-NTA agarose (Qiagen) for 30 min at 4 °C and 1,000 rpm. Eluted proteins were collected and combined with Laemmli buffer containing 2% (vol/vol) β-mercaptoethanol and 2 mM PMSF before analysis by SDS–PAGE and immunoblotting.

Double purification of Jac1–sfGFP stabilized TIM23 complexes was achieved by firstly isolating either Tim17[2xStrep]-containing complexes via Strep-Tactin XT magnetic beads (IBA Lifesciences GmbH) or [HisSumo*]Tim23-containing complexes using Ni-NTA agarose followed by a second purification using a GFP nanobody (GFP-Trap magnetic agarose; ChromoTek). Strep-tag purifications were performed as previously described[20]. Mitochondria were incubated in Strep solubilization buffer (20 mM HEPES/KOH, pH 7.4, 150 mM NaCl, 10% (vol/vol) glycerol, 0.1 mM EDTA, 1× protease inhibitor cocktail) containing 1% (wt/vol) digitonin end over end for 30 min at 4 °C. Insoluble material was removed by centrifugation (20,000g, 15 min, 4 °C), and mitochondrial lysates were incubated with Strep-Tactin XT magnetic beads for 2 h end over end at 4 °C. Bound protein complexes were washed five times in Strep solubilization buffer containing 0.3% (wt/vol) digitonin and eluted by incubation in 1× elution buffer BXT (IBA Lifesciences GmbH) containing 0.3% (wt/vol) digitonin, 10% (vol/vol) glycerol and 1× protease inhibitor cocktail for 1 h end over end at 4 °C. Eluates were collected and diluted 1:5 to a final concentration of 20 mM Tris-Cl, pH 7.4, 150 mM NaCl, 10% (vol/vol) glycerol, 0.2 mM EDTA, 10 mM biotin, 0.3% (wt/vol) digitonin and 1× protease inhibitor cocktail. For Ni-NTA affinity purification, [HisSumo*]Tim23 mitochondria were lysed in digitonin solubilization buffer containing 1% (wt/vol) digitonin and 10 mM imidazole end over end for 30 min at 4 °C. Clarified mitochondrial lysates were added to equilibrated Ni-NTA agarose and incubated for 1 h end over end at 4 °C. Bound proteins were washed (10 times) in digitonin solubilization buffer containing 0.1% (wt/vol) digitonin and 40 mM imidazole. Protein complexes were eluted with digitonin solubilization buffer containing 0.1% (wt/vol) digitonin and 250 mM imidazole. The elution was diluted to a final concentration of 20 mM Tris-Cl, pH 7.4, 150 mM NaCl, 10% (vol/vol) glycerol, 0.2 mM EDTA, 40 mM imidazole, 0.1% (wt/vol) digitonin and 1× protease inhibitor cocktail.

Diluted Strep or Ni-NTA eluates were incubated with equilibrated GFP nanobody magnetic agarose end over end for 1 h at 4 °C. GFP beads were washed three times with GFP solubilization buffer (10 mM Tris-Cl, pH 7.4, 150 mM NaCl, 0.5 mM EDTA, 10% (vol/vol) glycerol, 1× protease inhibitor cocktail) containing 0.3% (wt/vol) digitonin. Bound protein complexes were eluted with 0.2 M glycine, pH 2.5, neutralized with 1 M Tris-Cl, pH 9.4, and combined with Laemmli buffer containing 2 mM PMSF and 2% (vol/vol) β-mercaptoethanol before analysis by SDS–PAGE and immunoblotting.

Radiolabelled Cox5a–sfGFP-containing crosslinked products and oxidized Jac1–sfGFP-containing import intermediates were purified using GFP nanobody magnetic agarose. Mitochondria were solubilized in GFP solubilization buffer containing 1% (wt/vol) digitonin end over end for 30 min at 4 °C. Clarified lysate was added to equilibrated GFP agarose and incubated end over end for 1 h at 4 °C. GFP agarose was washed (three times), and bound proteins were eluted in 0.2 M glycine, pH 2.5. Eluates were neutralized with 1 M Tris-Cl, pH 9.4, combined with Laemmli buffer containing 2 mM PMSF and analysed by SDS–PAGE.

## Gel electrophoresis and western blotting
Native protein complexes were separated using BN–PAGE. Mitochondria were solubilized in digitonin buffer (20 mM Tris-HCl, pH 7.4, 50 mM NaCl, 0.1 mM EDTA, 10% (vol/vol) glycerol, 1 mM PMSF, 1% (wt/vol) digitonin (MATRIX BioScience)) for 15 min on ice. BN loading dye (0.5% (wt/vol) Coomassie blue G, 50 mM 6-aminocaproic acid, 10 mM Bis-Tris (Roth), pH 7.0) was added; samples were clarified (20,000g, 10 min, 4 °C), and supernatant was loaded onto a discontinuous gradient gel (6–16.5% or 3–14% acrylamide (48% (wt/vol) acrylamide, 1.5% (wt/vol) bisacrylamide)). Electrophoresis was performed in the presence of BN anode (50 mM Bis-Tris-HCl, pH 7.0) and BN cathode buffer (50 mM tricine, pH 7.0, 15 mM Bis-Tris with or without 0.02% (wt/vol) Coomassie blue G) at 4 °C, 100 V and 15–20 mA per gel overnight or at 600 V and 15–20 mA per gel for 3.5–5 h. For SDS–PAGE of radioactive proteins, samples were resuspended in 2× Laemmli buffer containing 2% (vol/vol) 2-mercaptoethanol and 2 mM PMSF and incubated for 15 min at 65 °C and 1,000 rpm. Denatured samples were loaded onto 10–15% polyacrylamide (Rotiphorese Gel 30 (30% acrylamide, 0.8% bisacrylamide (Roth)) gels and run in SDS running buffer (25 mM Tris-HCl, pH 8.8, 191 mM glycine (MP biomedicals), 1% (wt/vol) SDS) at 35 mA for 3–4 h. Tris-Tricine SDS–PAGE was performed with either straight 10% acrylamide or a 10–16.5% acrylamide gradient. Electrophoresis was performed using anode (0.2 M Tris-HCl, pH 8.9) and cathode buffers (0.1 M Tris, 0.1 M Tricine, 0.1% (wt/vol) SDS, pH 8.25) at 40–60 mA for 6 h or 15–20 mA overnight. For western blotting, proteins were subsequently transferred using a semi-dry system to PVDF membranes (Immobilon-P; Millipore) at 250 mA for 2.5 h in transfer buffer (20 mM Tris, 150 mM glycine, 0.02% (wt/vol) SDS, 20% (vol/vol) methanol). Membranes were stained (30% (vol/vol) ethanol, 10% (vol/vol) acetic acid, 0.2% (wt/vol) Coomassie R250), destained until protein bands became visible, cut and completely destained in 100% methanol. Membranes were washed in 20 mM Tris-HCl, pH 7.5, and 125 mM NaCl and then blocked for 1 h in blotting buffer (5% (wt/vol) skim milk in 20 mM Tris-HCl, pH 7.5, 125 mM NaCl). The membranes were incubated for 2 h at room temperature or overnight at 4 °C in primary antibody, washed and placed in secondary anti-rabbit IgG antibody (1:10,000 in blotting buffer; Sigma or Jackson ImmunoResearch Laboratories) or anti-mouse IgG antibody (1:2,000 in blotting buffer; Cell Signaling Technology) for 1–2 h at room temperature. Antibodies and dilutions are listed in Supplementary Table 4. After washing, membranes were incubated in ECL developing solution, and chemiluminescence signal was detected by an LAS-3000 or LAS-4000 camera system (Fujifilm) or X-ray films (Amersham Hyperfilm ECL; GE Healthcare). Gels separating radiolabelled precursor proteins were stained to compare protein loading (30% (vol/vol) ethanol, 10% (vol/vol) acetic acid, 1% (wt/vol) Coomassie R250), destained (30% (vol/vol) ethanol, 10% (vol/vol) acetic acid, 1% (wt/vol)), dried and exposed to PhosphoImager screens (GE Healthcare) followed by autoradiography using a Typhoon FLA 7000 PhosphoImager. Non-relevant gel lanes were excised digitally (indicated with a small gap).

## Mitochondrial membrane potential assessment
Mitochondrial membrane potential was assessed by fluorescence quenching. Isolated mitochondria were incubated in potential buffer (0.6 M sorbitol, 0.1% (wt/vol) BSA, 10 mM MgCl$_2$, 0.5 mM EDTA, 16 mM K$_2$HPO$_4$, 4 mM KH$_2$PO$_4$, pH 7.4) supplemented with 5 mM L-malate, 5 mM succinate and 2 μM 3,3′-dipropylthiadicarbocyanine iodide (Invitrogen). Absorption was measured on a Aminco Bowman II luminescence spectrophotometer (Thermo Fisher Corporation) at an excitation wavelength of 622 nm and emission wavelength of 670 nm until an equilibrium was reached, followed by the addition of 1 μM valinomycin to dissipate the membrane potential.

## Multiple sequence alignment of Tim17 family proteins
Multiple sequence alignment was performed using TMcoffee[60]. N- and C-terminal regions as well as the loop between TM1 and TM2 (denoted by dots in the sequence alignment between TM1 and TM2 in Extended Data Fig. 5a) were removed for simplification of the alignment figure. Location of transmembrane domain was assigned according to the cryo-EM structure of Tim22 (ref. 25) (Protein Data Bank (PDB) ID 6LO8) and following TMcoffee sequence alignment of *S. cerevisiae* Tim17 with Tim22.

## TIM23 complex modelling
SWISS-MODEL was used to model the full-length Tim17 monomer structure. The published structures of *S. cerevisiae* and *Homo sapiens*

Tim22 (PDB ID 6LO8 and PDB ID 7CGP) were used as model templates, and the resulting predicted structure was exported into UCSF ChimeraX (v.1.6.1)[61] for figure generation. Structural prediction of the Tim17–Tim23 heterodimer and heterotetramer as well as the interaction between further subunits of the TIM23 complex (Tim17–Tim23–Mgr2 and Tim17–Tim23–Mgr2–Pam17–Tim21–Tim50) was performed with ColabFold[47]. For these protein complex predictions, non-membrane predicted unstructured and partially disordered regions were excluded from modelling. Therefore, sequences used for modelling were as follows: Tim17 ($\Delta$143–158); Tim23 ($\Delta$1–90); Mgr2 ($\Delta$1–11 and $\Delta$86–113); Pam17 ($\Delta$1–47); Tim21 ($\Delta$1–60 and $\Delta$219–239); and Tim50 ($\Delta$1–101 and $\Delta$177–476). For structural prediction including cytochrome $b_2$ presequence, residues 1–40 were used. The highest ranked model was selected for analysis and figure generation. For the Tim17–Tim23 heterotetramer, the second-highest ranked model was additionally selected for comparison as it showed a different arrangement of the modelled subunits. Where relevant, residues were modified using the mutagenesis function in PyMOL (The PyMOL Molecular Graphics System, v.2.4; Schrödinger LLC), and the disulfide bond between $Tim17_{N16C}$ and $Tim23_{C98}$ formed. Predicted local distance difference test, predicted template modelling and interface predicted template modelling scores as well as predicted aligned error (PAE) plots were obtained from ColabFold predictions. PAE plots were visualized with PAE viewer[62].

## Statistics and reproducibility

Representative images are shown for in vitro imports of radiolabelled precursor proteins into isolated mitochondria; chemical crosslinking; BN electrophoresis; in vitro oxidation and reduction; yeast growth (WT and mutants); protein-level analysis and affinity purifications by immunodecoration; and assessment of the mitochondrial membrane potential of isolated mitochondria. The findings were confirmed by independent experiments (minimum numbers of independent experiments are in parentheses) for the following figures: Figs. 1c (2), 1d (3), 1f (3), 1g (3), 2b (2), 2c (2), 2d (2), 2e (2), 3b (2), 3c (3), 4a (2), 4b (2), 4c (2), 4d (2), 4e (2), 5b (2), 5c (2) and 5d (2) and Extended Data Figs. 1a (2), 1b (2), 1c (2), 1d (2), 2b (2), 2c (2), 2d (2), 2e (2), 2f (3), 3c (2), 3d (2), 3f (3), 5c (2), 6a (2), 6b (2), 6c (2), 6d (2), 6e (2), 6f (2), 6g (2), 6h (2), 6i (2), 7b (3), 7c (3), 7d (2), 7e (2), 7f (2), 7g (3), 7h (2), 7i (2), 8a (2), 8b (2), 8c (2), 8d (2), 8e (2), 8f (2), 8g (2) and 8h (2). The crucial experiments with phenotypes of Tim17 charge and hydrophilic mutants include biological replicates using different mitochondrial preparations (Figs. 1f,g, 3b,c and 4b,c, and Extended Data Figs. 2c,e,f, 6b and 7c–e,i), and the other experiments represent technical replicates.

## Miscellaneous

Data were collected by Typhoon FLA 7000 v.1.2 (build 1.2.1.93) and Image Reader FLA 7000 v.1.12 for autoradiography; Image Reader LAS-3000 v.2.21, Image Reader LAS-4000 v.2.1 and SilverFast (MicrotekSDK) v.6.6.1r7 for immunodecoration signals; AB2 Luminescence Spectrophotometer v.5.50 (Thermo Electron) for assessment of the mitochondrial membrane potential; CLARIOstar v.5.61 (BMG LABTECH) for monitoring yeast growth; Nanodrop ND-1000 v.3.5.2. (Coleman Technologies Inc. for Nanodrop Tech.) for spectroscopy; and Unicorn v.7.2. (General Electric Company) for usage of the Äkta pure system. Data were analysed and processed with PyMOL v.2.4.1 for introducing other amino acid residues to the structure of Tim17 and Tim23 and Affinity Photo v.1.10.6, ColabFold v.1.5.2, Geneious Prime v.2022.1.1, GraphPad Prism v.9.0.0 (121), ImageJ v.1.49v, Jalview v.2.11.2.7, Multi Gauge v.3.0 (Fujifilm), SnapGene viewer v.6.2.1, Adobe Illustrator 2023 v.27.6, Affinity Designer v.1.10.6, UCSF ChimeraX v.1.6.1 and CLARIOstar data analysis MARS v.3.41 (BMG LABTECH) for processing and figure preparation. Non-relevant lanes or yeast growth assays were removed by digital processing and are indicated by separating lines.

## Reporting summary

Further information on research design is available in the Nature Portfolio Reporting Summary linked to this article.

## Data availability

We used the following publicly available data: Tim22 structures from PDB (PDB 6LO8 and PDB 7CGP; https://www.rcsb.org); AlphaFold structural models (https://alphafold.ebi.ac.uk) of Tim17 (AF-P39515-F1) and Tim23 (AF-P32897-F1); and predicted interactions between Tim23 and Tim17 (https://doi.org/10.5452/ma-bak-cepc-0314) and between Tim17 and Mgr2 (https://doi.org/10.5452/ma-bak-cepc-0515). Uncropped gels/western blots are shown in Supplementary Fig. 1. Source data are provided with this paper.

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

**Acknowledgements** We thank P. Rehling, B. Guiard, N. Pfanner and M. Müller for material and discussion. Work included in this study was also performed in partial fulfilment of the requirements for the doctoral thesis of S.G.M. (University of Freiburg). This work was supported by the Deutsche Forschungsgemeinschaft (German Research Foundation; grant SFB 1381, project identification 403222702 (to H.-G.K., C.H. and N.W.); grant KO 2184/8, project identification 263098192 (to H.-G.K.); and grant WI 4506/1-1, project identification 406757425 (to N.W.)), Germany's Excellence Strategy (grant CIBSS–EXC-2189, project identification 390939984 (to C.H. and N.W.); grant FOR 2848, project identification 401510699 (to M.v.d.L.); and grant SFB 894, project identification 157660137 (to M.v.d.L.)) and the European Research Council (Consolidator Grant 648235 (to N.W.)). I.G. was supported by an Alexander von Humboldt Foundation Research Fellowship.

**Author contributions** M.v.d.L. and N.W. conceived and supervised the project. H.-G.K. was instrumental to establishing the initial site-specific crosslinking technology. L.F.F., J.D.B., I.G., C.S., N.Z., J.M.D. and L.B. performed cloning and analysis of yeast mutants. C.S. and H.B.H. performed cloning and isolation of sfGFP fusion proteins. S.J., K.N. and K.v.d.M. performed cloning of cytochrome $b_2$ cysteine variants. L.F.F., J.D.B., S.G.M., I.G., C.S., N.Z., S.J., K.N. and K.v.d.M. performed crosslinking. L.F.F., J.D.B., M.v.d.L. and N.W. analysed the experimental data with the co-authors. J.V.B., C.W. and C.H. performed and analysed the structural modelling. L.F.F. and J.D.B. prepared the figures with support of the co-authors. L.F.F., J.D.B. and N.W. wrote the manuscript with the support of all authors.

**Competing interests** The authors declare no competing interests.

**Additional information**
**Correspondence and requests for materials** should be addressed to Martin van der Laan or Nils Wiedemann.

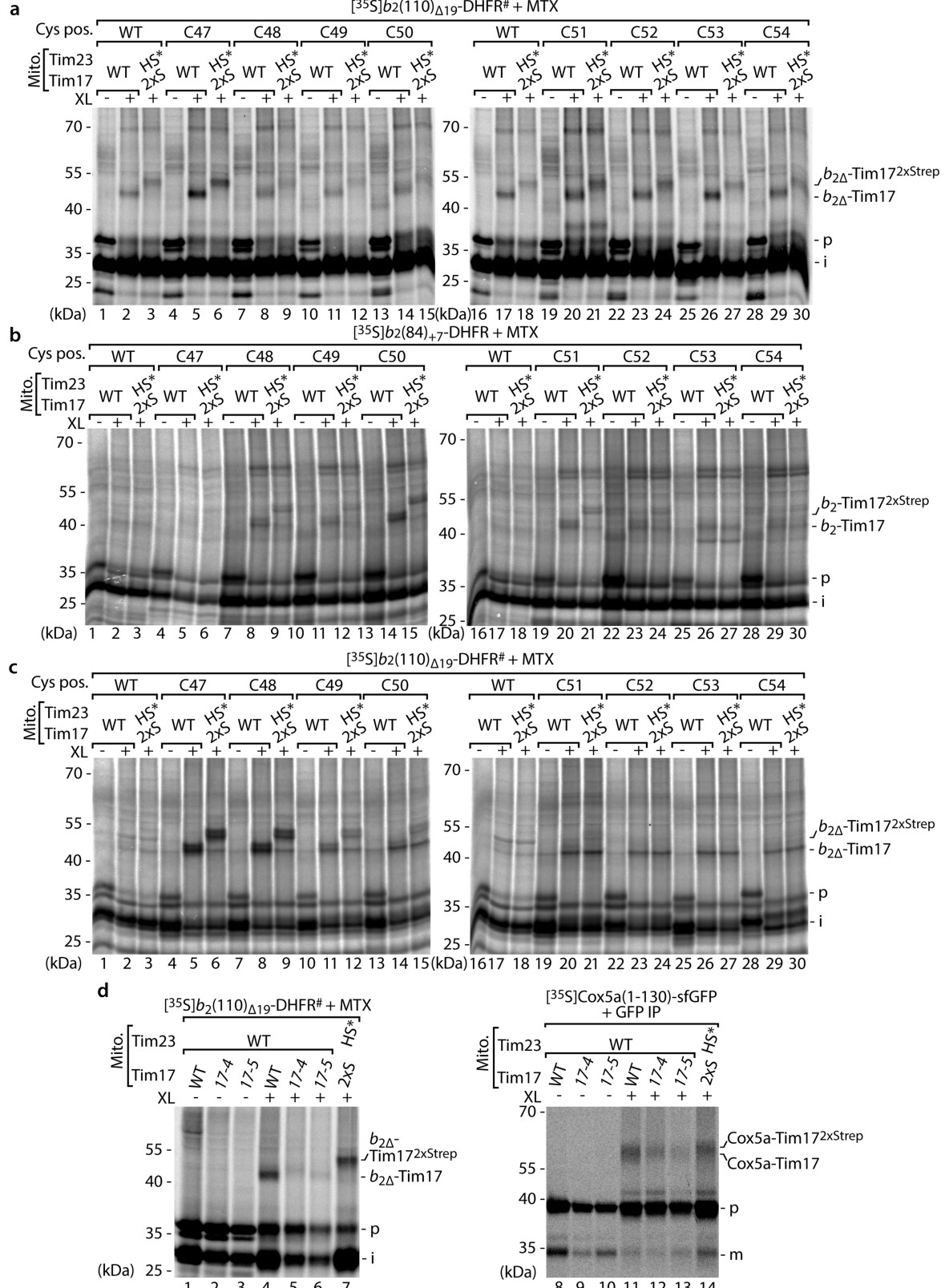

**Extended Data Fig. 1** | See next page for caption.

**Extended Data Fig. 1 | Arrested presequence precursors are in proximity to Tim17. a**–**c**, Import of $^{35}$S-labelled $b_2(110)_{\Delta 19}$-DHFR$^{\#}$ and $b_2(84)_{+7}$-DHFR with cysteine residues at the indicated positions (Cys pos.) in wildtype (WT) or Tim17$^{2xStrep}$ (2xS) and $^{HisSUMOstar}$Tim23 (HS*) mitochondria in the presence of methotrexate (MTX) followed by chemical crosslinking (XL) with m-maleimidobenzoyl-N-hydroxysuccinimide ester (MBS, **a**) or bismaleimidoethane (BMOE, **b** and **c**). Samples were analysed by SDS-PAGE and autoradiography. $b_{2\Delta}$-Tim17, $b_2(110)_{\Delta 19}$-DHFR$^{\#}$-Tim17 crosslinking product; $b_{2\Delta}$-Tim17$^{2xStrep}$, $b_2(110)_{\Delta 19}$-DHFR$^{\#}$-Tim17$^{2xStrep}$ crosslinking product; $b_2$-Tim17, $b_2(84)_{+7}$-DHFR-Tim17 crosslinking product; $b_2$-Tim17$^{2xStrep}$, $b_2(84)_{+7}$-DHFR-Tim17$^{2xStrep}$ crosslinking product; p, precursor; i, intermediate. **d**, Radiolabelled $b_2(110)_{\Delta 19}$-DHFR$^{\#}$ (lanes 1–7) or Cox5a(1–130)-sfGFP (lanes 8–14) were imported into Tim17 WT, *tim17-4*, *tim17-5* or Tim17$^{2xStrep}$ (2xS) and $^{HisSumo*}$Tim23 (HS*) mitochondria following heat shock. Crosslinked products were either directly analysed ($b_2(110)_{\Delta 19}$-DHFR$^{\#}$, lanes 1–7) or purified using a GFP nanobody (Cox5a-sfGFP, lanes 8–14) prior to analysis by SDS-PAGE and autoradiography. Cox5a-Tim17, Cox5a-sfGFP-Tim17 crosslinking product; Cox5a-Tim17$^{2xStrep}$, Cox5a-sfGFP-Tim17$^{2xStrep}$ crosslinking product; m, mature.

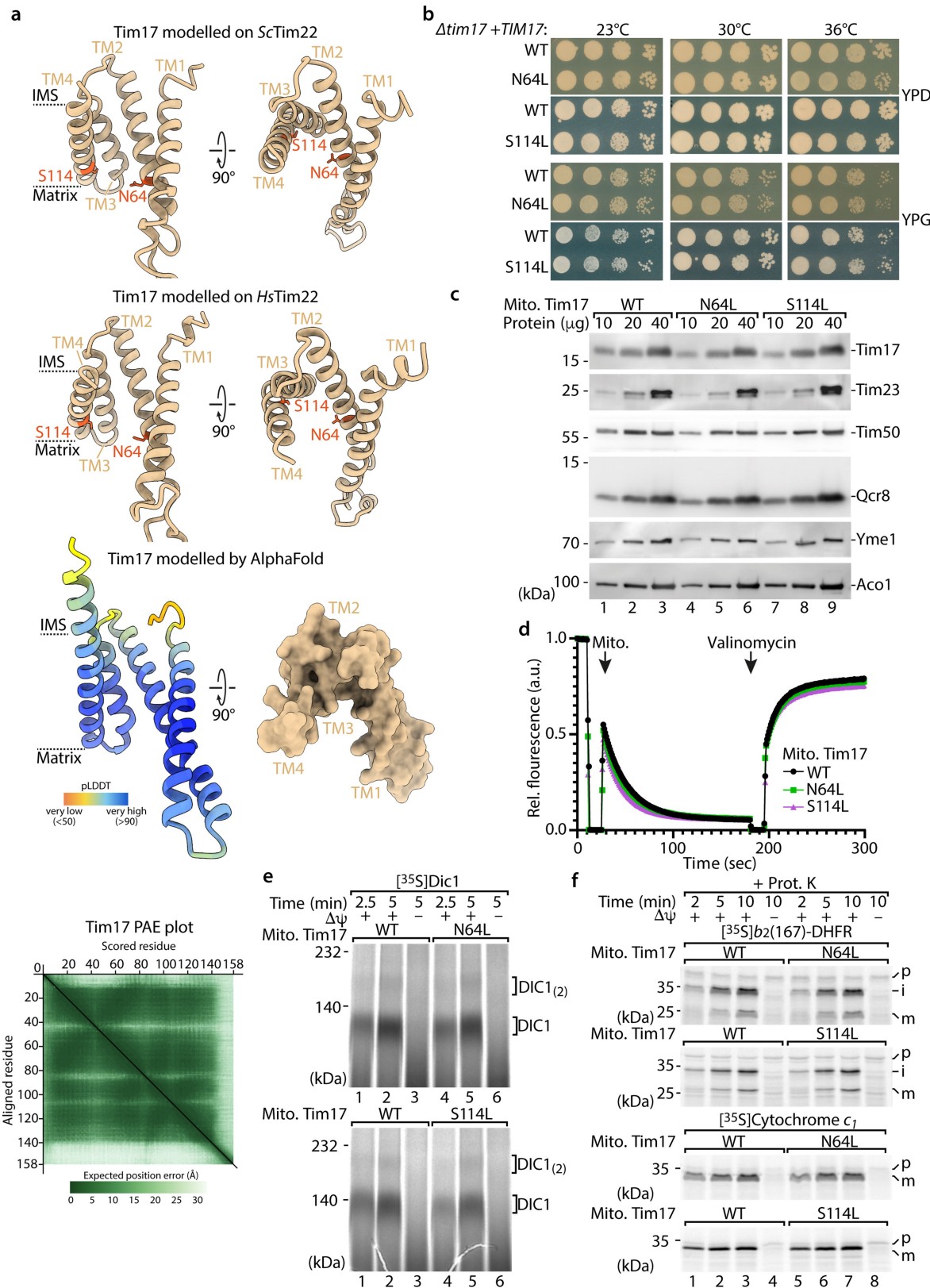

**Extended Data Fig. 2 |** See next page for caption.

**Extended Data Fig. 2 | Hydrophilic residues within the lateral Tim17 cavity are crucial for matrix translocation. a**, Tim17 (*S. cerevisiae*) modelled on the Tim22 (*S.c.*) structure (top; PDB ID 6LO8) and Tim22 (*H. sapiens*) structure (middle; PDB ID 7CGP). Dashed lines indicate the transmembrane (TM) segments of Tim17 (orange, asparagine 64 (N64), and serine (S114) residues). Predicted Local Distance Difference Test (pLDDT) score together with predicted aligned error plot (PAE) as well as surface representation of Tim17 as seen from the intermembrane space of the Tim17 AlphaFold model depicted in Fig. 1e (bottom). IMS, intermembrane space. **b**, Growth analysis of yeast strains expressing WT, Tim17[N64L] or Tim17[S114L] variants on agar with fermentable (dextrose/glucose, YPD) and non-fermentable (glycerol, YPG) medium at the indicated temperatures. **c**, Protein amounts of WT, Tim17[N64L] and Tim17[S114L]

mitochondria isolated from yeast cells grown on non-fermentable media (YPG) at 23 °C, analysed by SDS-PAGE and immunodecoration against the indicated antibodies. **d**, Membrane potential assessment of isolated WT, Tim17[N64L] and Tim17[S114L] mitochondria (from yeast cells grown in YPG at 23 °C). The membrane potential was assessed by fluorescence quenching using the potential sensitive dye 3,3′-dipropylthiadicarbocyanine iodide and subsequently dissipated by addition of valinomycin. **e**, Import of radiolabelled metabolite carrier protein dicarboxylate carrier 1 (Dic1) into isolated WT, Tim17[N64L] and Tim17[S114L] mitochondria, followed by blue native PAGE and autoradiography. $\Delta\psi$, membrane potential; $DIC1_{(2)}$, assembled $Dic_{(oligomer)}$. **f**, Import of radiolabelled $b_2(167)$-DHFR and cytochrome $c_1$ into isolated WT, Tim17[N64L] and Tim17[S114L] mitochondria followed by SDS-PAGE and autoradiography.

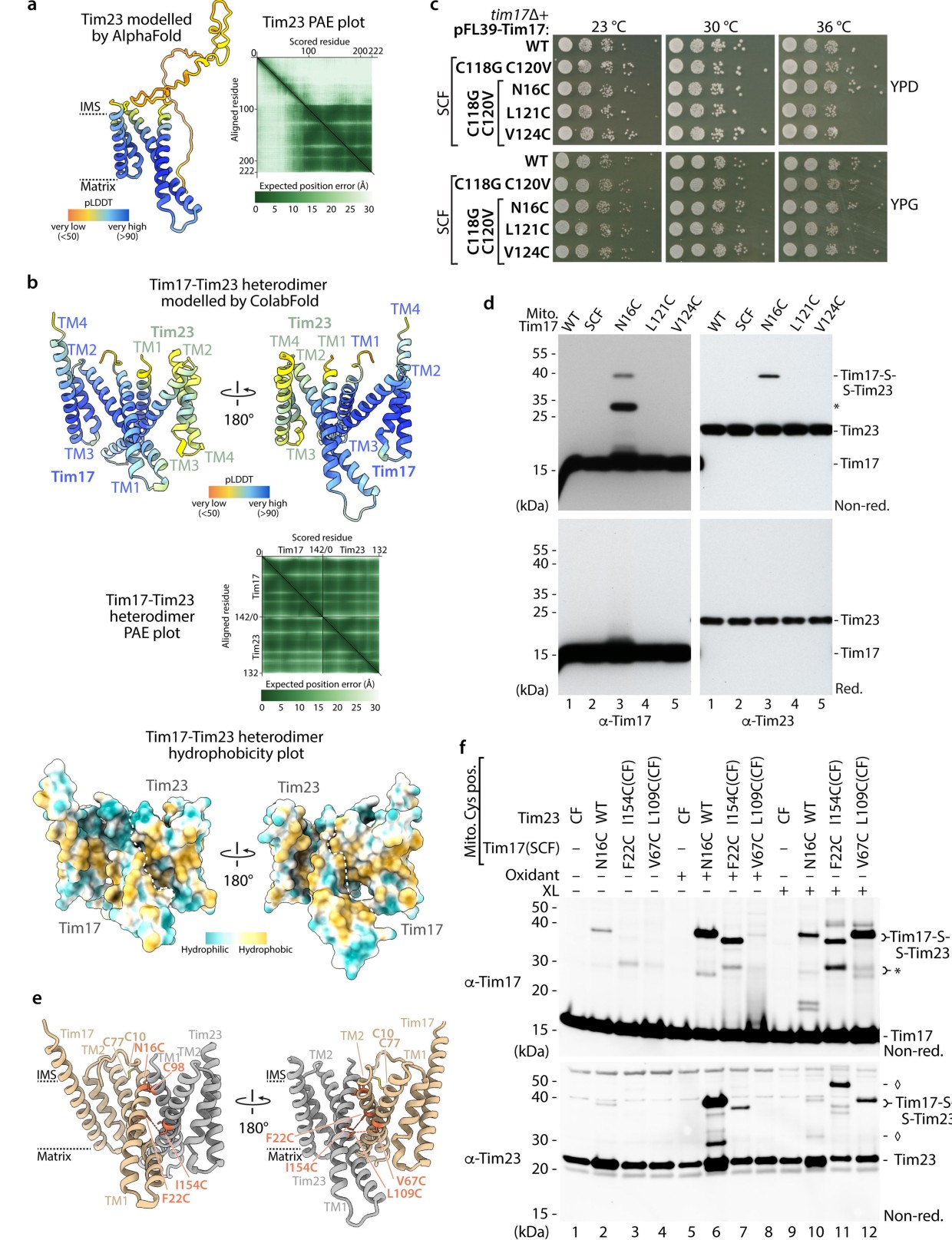

**Extended Data Fig. 3 |** See next page for caption.

**Extended Data Fig. 3 | Analysis of the Tim17-Tim23 back-to-back interaction with lateral cavities on opposing sides of the heterodimer. a**, Predicted Local Distance Difference Test (pLDDT) score plotted onto the Tim23 AlphaFold model (AF-P32897) alongside the associated predicted aligned error plot (PAE). **b**, Tim17-Tim23 heterodimer modelled by ColabFold. pLDDT score (upper), predicted aligned error (PAE, middle) and surface structure hydrophobicity (lower) plots. **c**, Growth analysis of yeast strains expressing WT or Tim17 cysteine mutant variants in the semi-cysteine-free background (SCF) as described for Extended Data Fig. 2b. **d**, Analysis of mitochondria isolated from yeast strains expressing WT, semi-cysteine-free Tim17 (SCF) and Tim17 cysteine variants by non-reducing (Non-red.) and reducing (Red.) SDS-PAGE and immunodecoration with the indicated antibodies. Tim17, Tim17 monomer; Tim23, Tim23 monomer; Tim17-S-S-Tim23, disulfide-linked Tim17 and Tim23; *, unidentified Tim17-specific conjugate **e**, ColabFold structural protein complex model of Tim17 (tan) and Tim23 (grey) heterodimer showing location of cysteine mutants (orange) at the dimer interface of the transmembrane regions of Tim17 and Tim23. Dotted lines, formation of disulphide bond or crosslinking product. **f**, Disulfide bond or crosslink formation between Tim17 and Tim23. Mitochondria isolated from yeast strains expressing Tim17 or Tim23 cysteine mutant variants in a semi-cysteine-free Tim17 (SCF) and Tim23 WT or cysteine-free (CF) background were left untreated or treated with oxidant (4-DPS; Oxidant) or crosslinker (BMOE; XL) followed by non-reducing SDS-PAGE and immunodecoration. ♦, unidentified Tim23-specific conjugate.

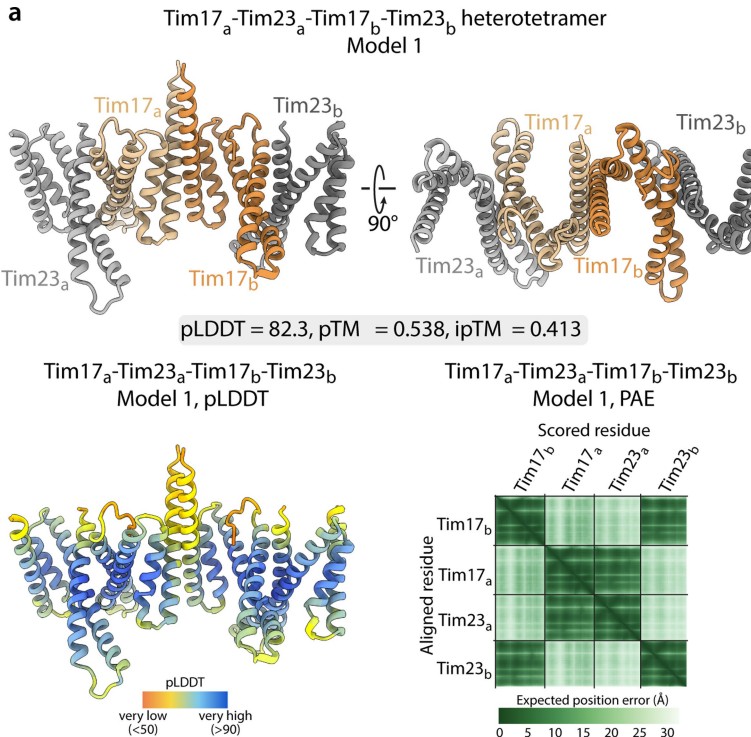

**a** Tim17$_a$-Tim23$_a$-Tim17$_b$-Tim23$_b$ heterotetramer
Model 1

pLDDT = 82.3, pTM = 0.538, ipTM = 0.413

Tim17$_a$-Tim23$_a$-Tim17$_b$-Tim23$_b$
Model 1, pLDDT

Tim17$_a$-Tim23$_a$-Tim17$_b$-Tim23$_b$
Model 1, PAE

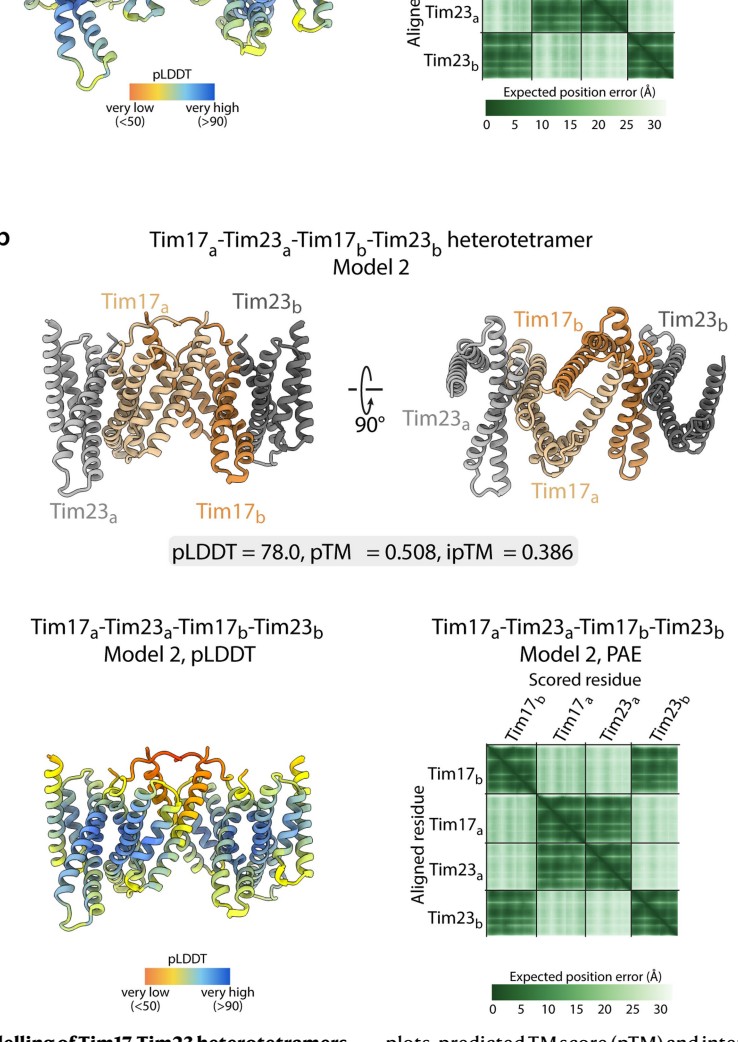

**b** Tim17$_a$-Tim23$_a$-Tim17$_b$-Tim23$_b$ heterotetramer
Model 2

pLDDT = 78.0, pTM = 0.508, ipTM = 0.386

Tim17$_a$-Tim23$_a$-Tim17$_b$-Tim23$_b$
Model 2, pLDDT

Tim17$_a$-Tim23$_a$-Tim17$_b$-Tim23$_b$
Model 2, PAE

**Extended Data Fig. 4 | Structural modelling of Tim17-Tim23 heterotetramers. a,b**, Representative ColabFold structural protein complex models of Tim17 (a/b: tan/orange) and Tim23 (a/b: grey/dark grey) within putative weakly predicted heterotetrameric arrangements with associated pLDDT scores, PAE plots, predicted TM score (pTM) and interface pTM (ipTM) scores. These two distinct models predicted by ColabFold represent those with the highest ranking.

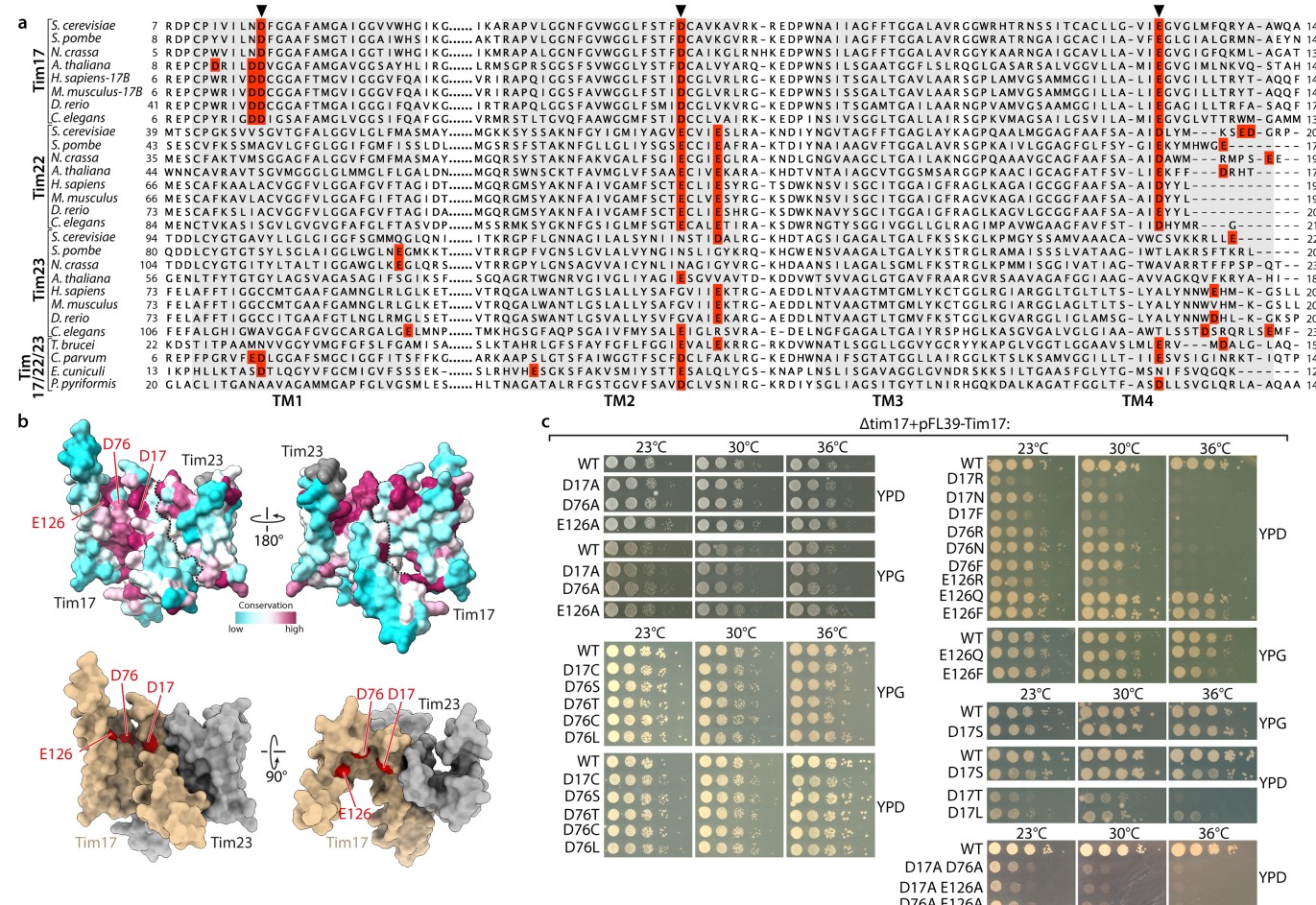

**Extended Data Fig. 5 | Sequence alignment of Tim17 protein family with conserved negative charges within the TMDs and growth phenotype of corresponding *tim17* mutants. a**, Sequence alignment of Tim17 from *Saccharomyces cerevisiae*, *Schizosaccharomyces pombe*, *Neurospora crassa*, *Caenorhabditis elegans*, *Drosophila melongaster*, *Danio rerio*, *Homo sapiens*, *Mus musculus* and *Arabidopsis thaliana* (paralog 1) generated by TMcoffee[60]. Predicted TMDs were assigned following sequence alignment of *S. c.* Tim17 with *S. c.* Tim22 and according to the cyro-EM structure of Tim22 (6LO8)[25]. Grey boxes, predicted transmembrane (TM) domains; Red, highly conserved negatively charged residues within predicted TM domains; Arrowheads,

negatively charged residues analysed in this study. **b**, Tim17 and Tim23 amino acid sequence conservation mapped onto the surface of the ColabFold predicted Tim17-Tim23 heterodimer (upper panel). Space filling model of the Tim17-Tim23 heterodimer with the conserved negatively charged acidic patch formed by D17, D76 and E126 indicated (lower panel). **c**, Growth analysis of WT and *tim17* mutants in *tim17*Δ background related to Extended Data Table 1. Yeast strains expressing WT or the indicated *tim17* mutant variant were grown on non-fermentable (glycerol, YPG) or fermentable (dextrose/glucose, YPD) agar medium at the indicated temperatures.

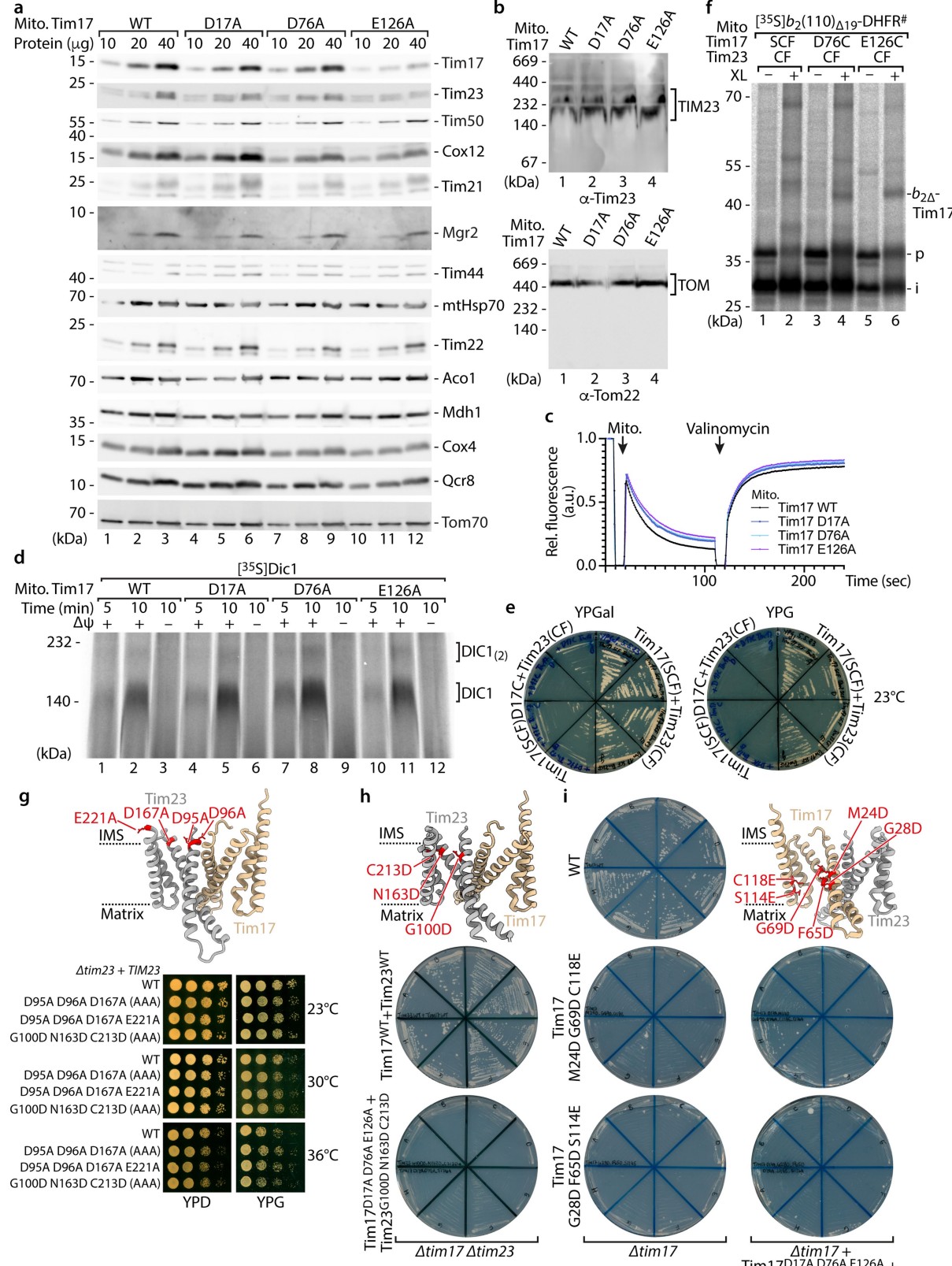

**Extended Data Fig. 6** | See next page for caption.

**Extended Data Fig. 6 | Tim17$^{D17A}$ and Tim17$^{D76A}$ mitochondria have no obvious secondary defects and introduction of equivalent negative charges into Tim23 or lower parts of the lateral cavity of Tim17. a**, Protein amounts of WT, Tim17$^{D17A}$, Tim17$^{D76A}$ and Tim17$^{E126A}$ mitochondria isolated from yeast cells grown in YPG at 23 °C, followed by SDS-PAGE and immunodecoration with the indicated antibodies. **b**, Protein complexes of isolated Tim17 WT and mutant yeast mitochondria analysed by BN-PAGE and immunodecoration with the indicated antibodies. TIM23, presequence translocase of the inner membrane; TOM, translocase of the outer mitochondrial membrane. **c**, Assessment of the mitochondrial membrane potential of isolated mitochondria as described in Extended Data Fig. 2d. **d**, Import of radiolabelled metabolite carrier protein Dic1 into isolated WT and *tim17* mutant mitochondria, followed by BN-PAGE and autoradiography. **e**, Growth analysis of *tim17 tim23* mutant yeast strains expressing Tim17$^{SCF}$+Tim23$^{CF}$ or Tim17$^{D17C}$+Tim23$^{CF}$ on agar media containing galactose (YPGal) and glycerol (YPG) at 23 °C. **f**, Import of $^{35}$S-labelled

$b_2(110)_{\Delta 19}$-DHFR$^{\#}$ into Tim17 WT or cysteine mutant mitochondria in the presence of MTX followed by chemical crosslinking (MBS; XL), SDS-PAGE and autoradiography. **g**, ColabFold structural protein complex model of *S. c.* Tim17 (tan) and Tim23 (grey) heterodimer with the location of the negatively charged residues (red) within the transmembrane domains and adjacent segments of Tim23 (upper panel). Growth analysis of *tim23* mutant yeast strains expressing Tim23 WT or negative charge mutant variants as for Extended Data Fig. 2b (lower panel). AAA, Tim23 D95A D96A D167A. **h, i**, ColabFold structural protein complex model as for **g** with the location of residues equivalent to Tim17 D17, D76 and E126 that were introduced into the transmembrane regions of Tim23 (**h**) or locations with equivalent charged residues, located lower within the lateral cavity of Tim17, were introduced (**i**) (red; upper panels). Growth phenotypes of *tim17 tim23* mutant yeast strains in the *tim17Δtim23Δ* background (**h**) or *tim17* mutants in the *tim17Δ* and *tim17Δ*, Tim17$^{D17A\_D76A\_E126A}$ background (**i**) on 5-FOA medium at 23 °C (lower panel).

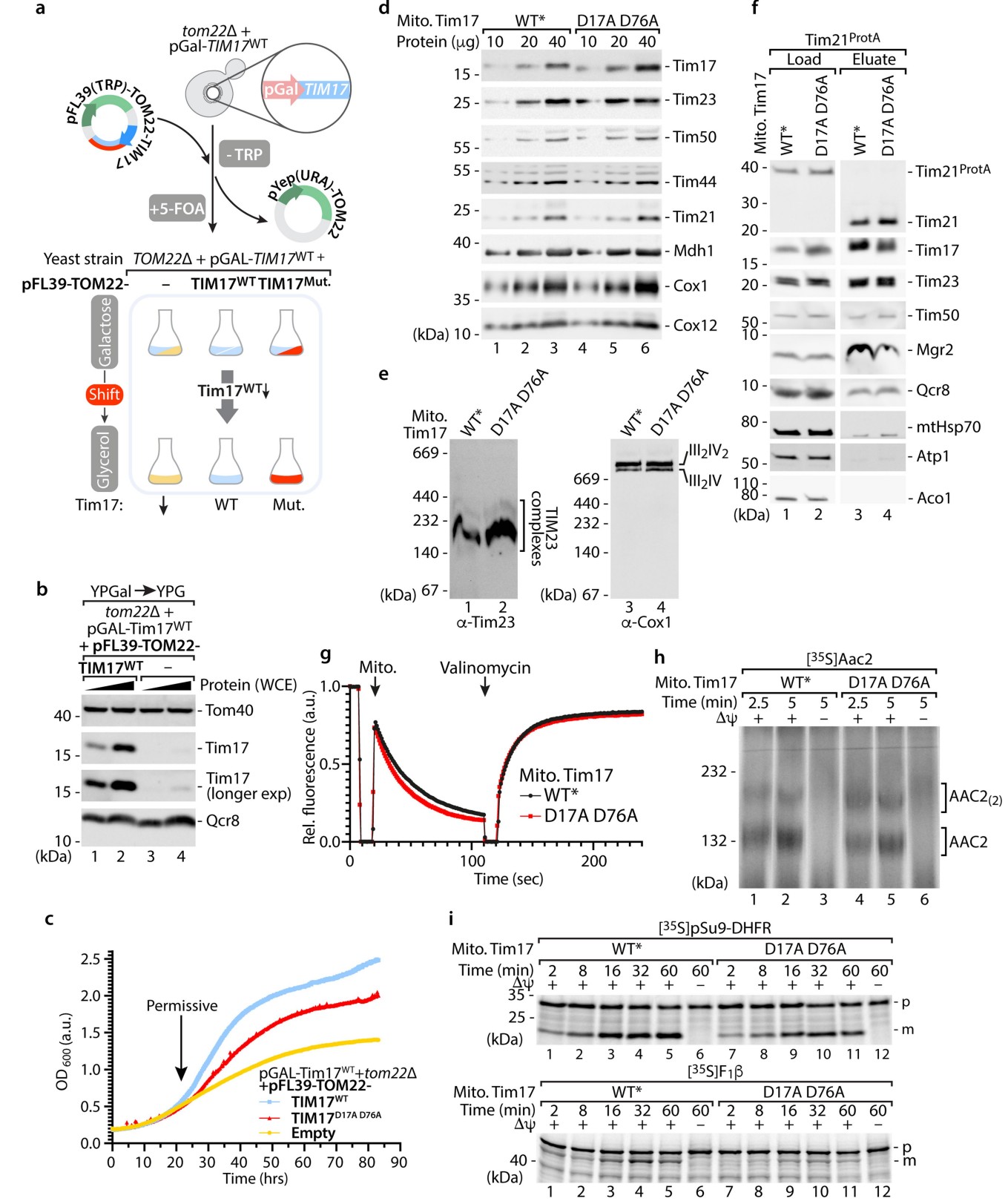

**Extended Data Fig. 7** | See next page for caption.

**Extended Data Fig. 7 | Yeast expressing Tim17$^{D17A\_D76A}$ and analysis of Tim17$^{D17A\_D76A}$ mitochondria. a**, Graphical depiction of the approach to generate Tim17 mutants by placing chromosomal *TIM17* under the control of the galactose promotor (pGAL-*TIM17*) in a *TOM22* deletion background. This strain was transformed with single copy **pFL(*TRP*)** plasmids, encoding WT *TOM22* and the desired *TIM17* alleles. After elimination of the ***pYEP(URA)-TOM22*** cover plasmid by 5-Fluoroorotic acid (5-FOA) treatment, the **pFL** plasmid is maintained, as the yeast cells require the *TOM22* gene for viability. pGal, galactose inducible promoter; TRP, tryptophan; URA, uracil; Mut., *tim17* mutant. **b**, Analysis of Tim17$^{WT}$ protein depletion by shifting the pGAL-TIM17$^{WT}$ strain from galactose to fermentable glycerol medium for 21 h. Protein levels in yeast whole cell extracts (WCE) were analysed by SDS-PAGE and immunodecoration. **c**, Growth analysis of *tom22*Δ, pGal-*TIM17*$^{WT}$ + **pFL39-*TOM22*** yeast strains expressing Tim17$^{WT}$ (blue), Tim17$^{D17A\_D76A}$ (red) or no Tim17/'empty' (yellow) in glycerol-containing media at 25 °C. Arrow indicates the time before the growth of the strain expressing Tim17$^{D17A\_D76A}$ is affected compared to the WT strain. **d**, Protein amounts of WT* and Tim17$^{D17A\_D76A}$ mitochondria isolated from yeast cells shifted to non-fermentable media (YPG) at 23 °C, analysed by SDS-PAGE and immunodecoration against the indicated antibodies. WT*, WT strain with Tim17 protein levels comparable to Tim17$^{D17A\_D76A}$ mutant **e**, Protein complexes of isolated Tim17 WT and Tim17$^{D17A\_D76A}$ mutant yeast mitochondria were analysed by BN-PAGE and immunodecoration. III$_2$IV, III$_2$IV$_2$, respiratory chain supercomplexes. **f**, TIM23 complex isolation from digitonin-solubilised Tim17$^{WT}$ + Tim21$^{ProtA}$ or Tim17$^{D17A\_D76A}$ + Tim21$^{ProtA}$ mitochondria. Bound protein complexes were analysed by SDS-PAGE and immunodecoration with the indicated antibodies. ProtA, Protein A; Load, 1%; Eluate 100%. **g**, Mitochondrial membrane potential of WT and Tim17$^{D17A\_D76A}$ mitochondria isolated from yeast cells grown in non-fermentable glycerol medium. Assessment of the membrane potential of isolated mitochondria as described in Extended Data Fig. 2d. **h**, Import of radiolabelled metabolite carrier ADP/ATP carrier (Aac2) into isolated WT and Tim17$^{D17A\_D76A}$ mitochondria followed by BN-PAGE and autoradiography. AAC$_{(2)}$, assembled ADP/ATP carrier$_{(oligomer)}$. **i**, Import of radiolabelled pSu9-DHFR (top) and F$_1$β (Atp2, bottom) into WT and Tim17$^{D17A\_D76A}$ mitochondria isolated as described in Extended Data Fig. 2f.

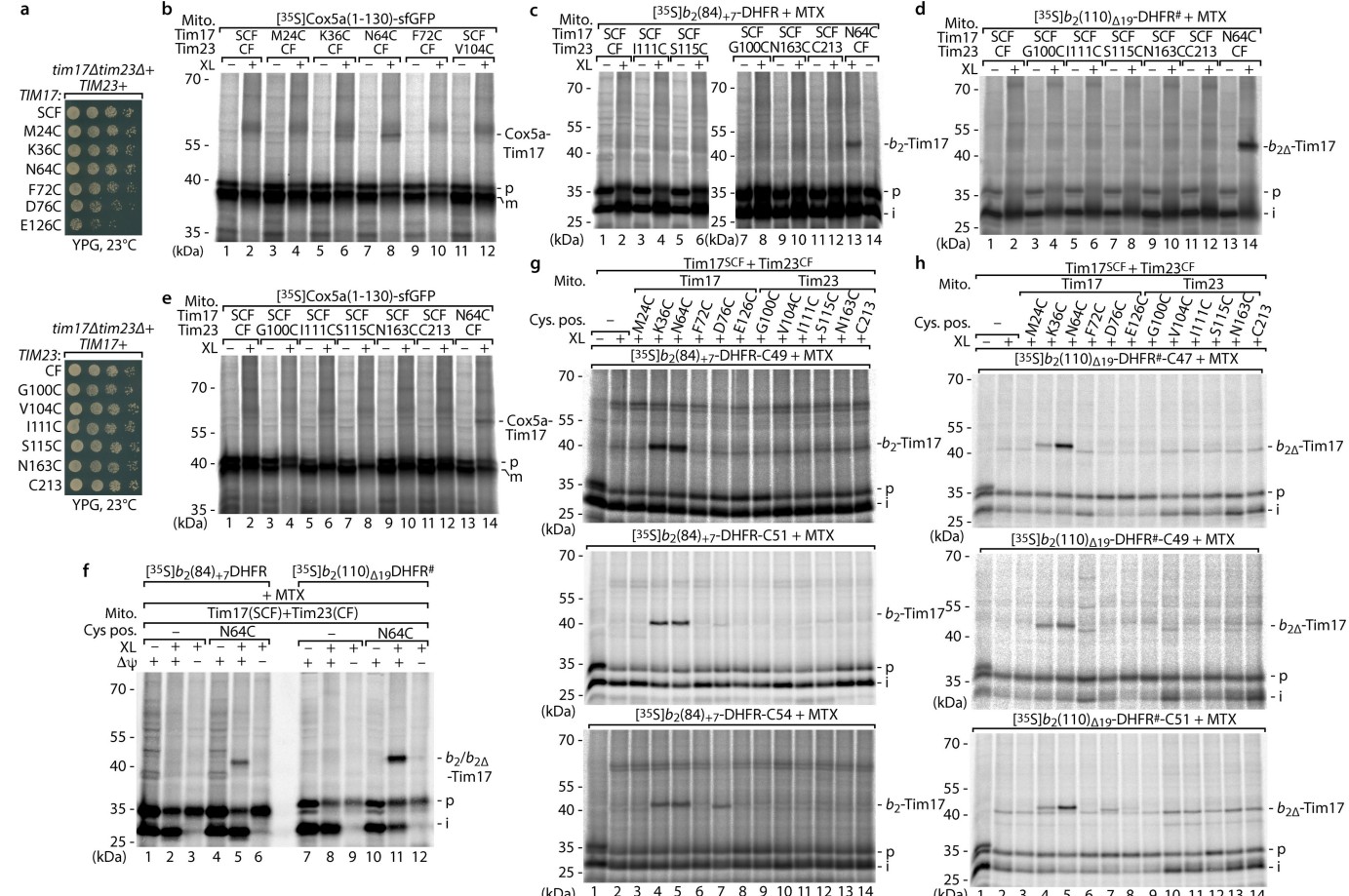

**Extended Data Fig. 8 | Interaction of precursor proteins with the lateral Tim17 cavity is membrane potential-dependent. a**, Growth phenotypes of the indicated *tim17* (top panel) or *tim23* (bottom panel) cysteine mutants in the *tim17Δtim23Δ* background on YPG agar at 23 °C. **b-e**, Arrest of [35S]-labelled Cox5a(1–130)-sfGFP (**b** and **e**), $b_2(84)_{+7}$-DHFR (**c**) or $b_2(110)_{Δ19}$-DHFR[#] (**d**) into Tim17[SCF] and Tim23[CF] mitochondria with cysteine residues introduced at the indicated positions. Chemical crosslinking (XL) was performed with MBS followed by SDS-PAGE and autoradiography. **f**, Methotrexate (MTX) preincubated, radiolabelled $b_2(84)_{+7}$-DHFR or $b_2(110)_{Δ19}$-DHFR[#] was imported

into Tim17[SCF] or Tim17[SCF_N16C] mitochondria in the presence or absence of a membrane potential (Δψ) across the inner membrane followed by chemical crosslinking (XL) with MBS. Samples were subsequently analysed by SDS-PAGE and autoradiography. **g,h**, Radiolabelled $b_2(84)_{+7}$-DHFR (g) or $b_2(110)_{Δ19}$-DHFR[#] (h) constructs with cysteine residues at the indicated positions were imported into Tim17[SCF]+Tim23[CF] control or mutant mitochondria with cysteine residues at the indicated position (Cys. pos.) within the transmembrane cavity. Import was followed by chemical crosslinking (XL) with BMOE, SDS-PAGE and autoradiography.

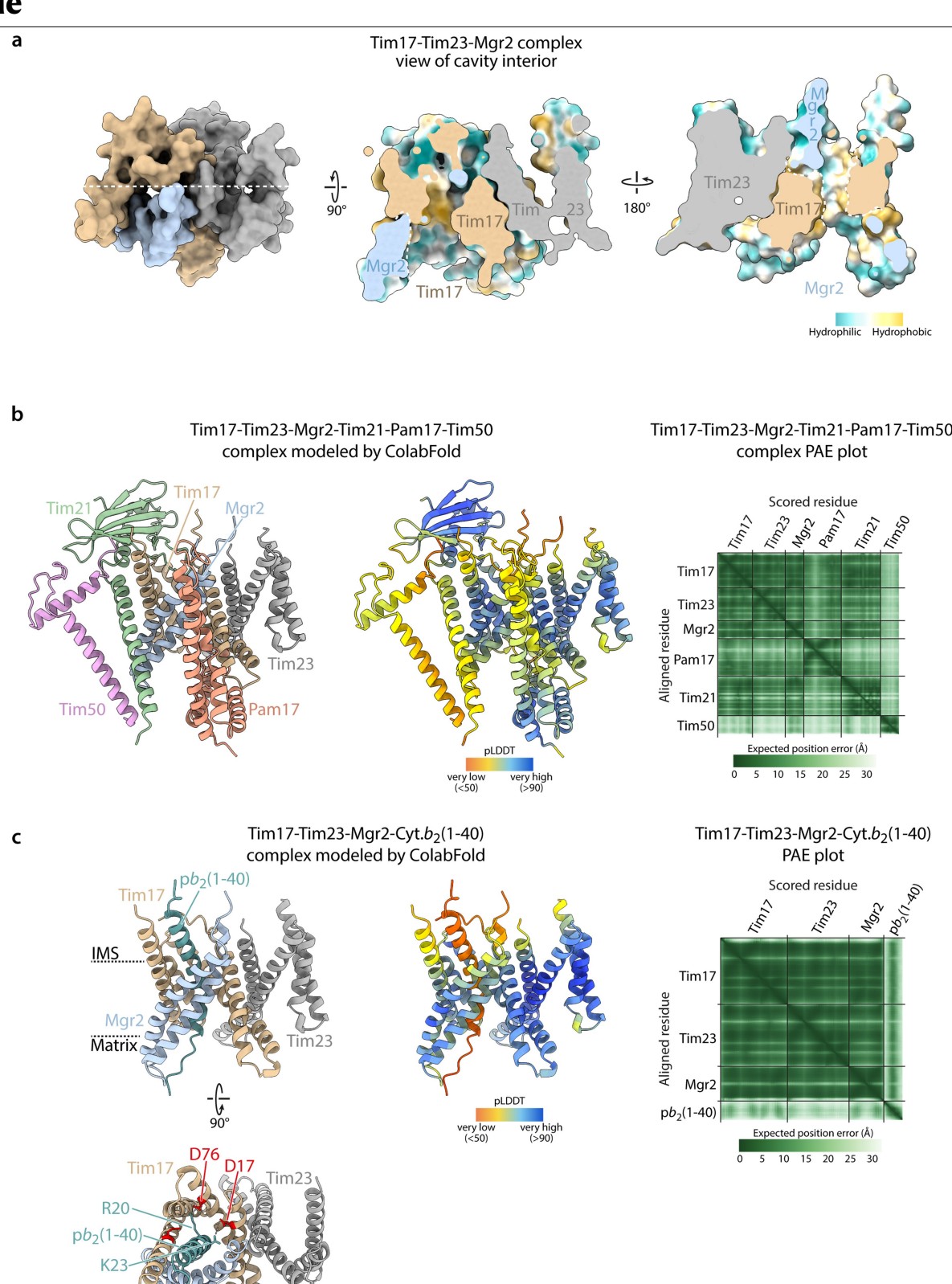

**Extended Data Fig. 9** | See next page for caption.

**Extended Data Fig. 9 | TIM23 complex modelling. a**, ColabFold structural protein complex prediction of the interaction between *S. c.* Tim17, Tim23 and Mgr2 as depicted in Fig. 6a (left) and displayed 'cut-open' with hydrophilic and hydrophobic residues mapped onto the protein surface (middle and right). **b**, ColabFold structural prediction of the complex formed by *S. c.* Tim17 (tan), Tim23 (grey), Mgr2 (blue), Pam17 (salmon), Tim21 (green) and Tim50 (pink) with associated pLDDT scores and PAE plots. The TIM23 subunits with transmembrane segments Mgr2, Pam17, Tim21 and Tim50 are all predicted to associate at the Tim17 side of the Tim17-Tim23 heterodimer. Compared to the high confidence prediction that Mgr2 can directly associate to the Tim17 lateral cavity; all the other subunits have lower confidence predictions and are more peripheral compared to Mgr2. **c**, ColabFold structural protein complex prediction of the heterotrimer formed by *S. c.* Tim17 (tan), Tim23 (grey), Mgr2 (blue) and residues 1–40 of cytochrome $b_2$ (p$b_2$) including the presequence with associated pLDDT scores and PAE plots. Dotted lines, putative interactions between Tim17 D17 and D76 and indicated positively charged residues of p$b_2$; red, Tim17 *TIS* residues.

**Extended Data Table 1 | Summary of growth phenotypes of Tim17 mutants**

| pFL39-Tim17 | Δtim17 YPD 23°C | YPD 30°C | YPD 36°C | YPG 30°C | YPG 36°C |
|---|---|---|---|---|---|
| D17A | + | + | + | + | + |
| D17S | + | + | + | + | + |
| D17L | – | – | – | † | |
| D17N | – | – | † | † | |
| D17T | – | – | † | † | |
| D17F | – | – | † | † | |
| D17R | – | – | † | † | |
| D76A | + | + | + | + | + |
| D76S | + | + | + | – | – |
| D76T | + | + | – | – | – |
| D76L | + | + | – | – | – |
| D76F | – | – | † | † | |
| D76N | – | – | † | † | |
| D76R | – | – | † | † | |
| E126A | + | + | + | + | + |
| E126Q | + | + | + | + | + |
| E126F | + | + | + | + | – |
| E126R | – | – | † | † | |
| D17A D76A | – | – | † | † | |
| D17A E126A | – | – | † | † | |
| D76A E126A | – | – | † | † | |
| Triple (AAA) | | † | | † | |

Summary of growth phenotypes of Tim17 single, double and triple point mutants of the negatively charged transmembrane domain residues on fermentable (glucose, YPD) and non-fermentable (glycerol, YPG) medium at the indicated temperatures. Triple (AAA), Tim17$^{D17A\_D76A\_E126A}$; Green, viable; Pink, moderate growth defect; Red, severe growth defect; Black, no growth (related to Extended Data Fig. 5c).

Nils Wiedmann

# Reporting Summary

## Statistics

For all statistical analyses, confirm that the following items are present in the figure legend, table legend, main text, or Methods section.

| n/a | Confirmed | |
|---|---|---|
| ☒ | ☐ | The exact sample size (*n*) for each experimental group/condition, given as a discrete number and unit of measurement |
| ☒ | ☐ | A statement on whether measurements were taken from distinct samples or whether the same sample was measured repeatedly |
| ☒ | ☐ | The statistical test(s) used AND whether they are one- or two-sided *Only common tests should be described solely by name; describe more complex techniques in the Methods section.* |
| ☒ | ☐ | A description of all covariates tested |
| ☒ | ☐ | A description of any assumptions or corrections, such as tests of normality and adjustment for multiple comparisons |
| ☒ | ☐ | A full description of the statistical parameters including central tendency (e.g. means) or other basic estimates (e.g. regression coefficient) AND variation (e.g. standard deviation) or associated estimates of uncertainty (e.g. confidence intervals) |
| ☒ | ☐ | For null hypothesis testing, the test statistic (e.g. $F$, $t$, $r$) with confidence intervals, effect sizes, degrees of freedom and $P$ value noted *Give P values as exact values whenever suitable.* |
| ☒ | ☐ | For Bayesian analysis, information on the choice of priors and Markov chain Monte Carlo settings |
| ☒ | ☐ | For hierarchical and complex designs, identification of the appropriate level for tests and full reporting of outcomes |
| ☒ | ☐ | Estimates of effect sizes (e.g. Cohen's *d*, Pearson's *r*), indicating how they were calculated |

*Our web collection on statistics for biologists contains articles on many of the points above.*

## Software and code

Policy information about availability of computer code

| Data collection | Typhoon FLA-7000 ver. 1.2 (build 1.2.1.93), Image Reader FLA-7000 ver. 1.12 for autoradiography; Image Reader LAS-3000 ver. 2.21, Image Reader LAS-4000 ver. 2.1, SilverFast (MicrotekSDK) ver. 6.6.1r7 for immunodecoration signals; AB2 Luminescence Spectrophotometer ver. 5.50 (Thermo Electron) for assessment of the mitochondrial membrane potential; CLARIOstar ver. 5.61 (BMG LABTECH) for monitoring yeast growth; Nanodrop ND-1000 ver. 3.5.2. (Coleman Technologies Inc. for Nanodrop Tech.) for spectroscopy; Unicorn ver. 7.2. (General Electric Company) for usage of the Äkta pureTM system. |
|---|---|
| Data analysis | PyMOL ver. 2.4.1 for introducing other amino acid residues to the structure of Tim17 and Tim23; Affinity Photo ver. 1.10.6., ColabFold ver. 1.5.2, Geneious Prime ver. 2022.1.1, GraphPad Prism ver. 9.0.0 (121), ImageJ ver. 1.49v, Jalview ver. 2.11.2.7., Multi Gauge ver. 3.0 (Fujifilm), SnapGene viewer ver. 6.2.1, Adobe Illustrator 2023 ver. 27.6., Affinity Designer ver. 1.10.6., UCSF ChimeraX ver. 1.6.1., CLARIOstar data analysis MARS ver. 3.41 (BMG LABTECH) for processing and figure preparation. |

For manuscripts utilizing custom algorithms or software that are central to the research but not yet described in published literature, software must be made available to editors and reviewers. We strongly encourage code deposition in a community repository (e.g. GitHub). See the Nature Portfolio guidelines for submitting code & software for further information.

## Data

Policy information about availability of data

All manuscripts must include a data availability statement. This statement should provide the following information, where applicable:

- Accession codes, unique identifiers, or web links for publicly available datasets
- A description of any restrictions on data availability
- For clinical datasets or third party data, please ensure that the statement adheres to our policy

We employed the following publicly available data: Tim22 structures from (PDB ID 6LO8 and PDB ID 7CGP; https://www.rcsb.org); AlphaFold structural models (https://alphafold.ebi.ac.uk) of Tim17 (AF-P39515-F1) and Tim23 (AF-P32897-F1); predicted interaction between Tim23 and Tim17 (doi:10.5452/ma-bak-cepc-0314) and between Tim17 and Mgr2 (doi:10.5452/ma-bak-cepc-0515). Uncropped gels, blots and source data are shown in Supplementary Fig. 1 and Supplementary Table 5.

## Human research participants

Policy information about studies involving human research participants and Sex and Gender in Research.

| Reporting on sex and gender | N/A |
| --- | --- |
| Population characteristics | N/A |
| Recruitment | N/A |
| Ethics oversight | N/A |

Note that full information on the approval of the study protocol must also be provided in the manuscript.

# Field-specific reporting

Please select the one below that is the best fit for your research. If you are not sure, read the appropriate sections before making your selection.

☒ Life sciences ☐ Behavioural & social sciences ☐ Ecological, evolutionary & environmental sciences

For a reference copy of the document with all sections, see nature.com/documents/nr-reporting-summary-flat.pdf

# Life sciences study design

All studies must disclose on these points even when the disclosure is negative.

| Sample size | Sample sizes used for biochemical experiments were selected based on previous experiences with specific types of experiments like the amount of mitochondria used for protein level analysis, in vitro import and pulldown experiments (Ieva et al., 2014, doi:10.1016/j.molcel.2014.10.010; Weinhäupl et al., 2018, doi:10.1016/j.cell.2018.10.039; Höhr et al., 2018, doi:10.1126/science.aah6834; Takeda et al., 2021, doi:10.1038/s41586-020-03113-7). According to this the required amount of yeast cells or mitochondria was selected for each experiment. For experiments with modifications compared to previous types, several runs were performed to determine an optimal sample size. The sample sizes for the experiments are stated in the Methods section. |
| --- | --- |
| Data exclusions | All relevant data shown. No data were excluded from the analysis. |
| Replication | Representative images are shown for in vitro imports of radiolabelled precursor proteins into isolated mitochondria, chemical crosslinking, blue native electrophoresis, in vitro oxidation and reduction, yeast growth (wild-type and mutants), protein level analysis and affinity purifications by immunodecoration, assessment of the mitochondrial membrane potential of isolated mitochondria. The findings were confirmed by independent experiments (minimum number of independent experiments in parentheses) for the following figures 1c (2), 1d (3), 1f (3), 1g (3), 2b (2), 2c (2), 2d (2), 2e (2), 3b (2), 3c (3), 4a (2), 4b (2), 4c (2), 4d (2), 4e (2), 5b (2), 5c (2), 5d (2) and Extended Data figures ED1a (2), ED1b (2), ED1c (2), ED1d (2), ED2b (2), ED2c (2), ED2d (2), ED2e (2), ED2f (3), ED3c (2), ED3d (2), ED3f (3), ED5c (2), ED6a (2), ED6b (2), ED6c (2), ED6d (2), ED6e (2), ED6f (2), ED6g (2), ED6h (2), ED6i (2), ED7b (3), ED7c (3), ED7d (2), ED7e (2), ED7f (2), ED7g (3), ED7h (2), ED7i (2), ED8a (2), ED8b (2), ED8c (2), ED8d (2), ED8e (2), ED8f (2), ED8g (2), ED8h (2). The crucial experiments with phenotypes of Tim17 charge and hydrophilic mutants include biological replicates using different mitochondrial preparations (Fig. 1f-g; Fig. 3b-c; Fig. 4b-c; Extended data Fig. 2c, e-f; Extended data Fig. 6b; Extended data Fig. 7c-e and i), the other experiments represent technical replicates. |
| Randomization | All yeast clones for growth tests, mitochondrial isolations and biochemical experiments were selected randomly. The experiments employing isolated mitochondria were not randomized. All samples in one experiment were processed in parallel with the same buffers and under the same conditions. |
| Blinding | Blinding was not performed. The yeast strains had to be validated before they were used for experiments. |

# Reporting for specific materials, systems and methods

We require information from authors about some types of materials, experimental systems and methods used in many studies. Here, indicate whether each material, system or method listed is relevant to your study. If you are not sure if a list item applies to your research, read the appropriate section before selecting a response.

## Materials & experimental systems

| n/a | Involved in the study |
|-----|----------------------|
| ☐ | ☒ Antibodies |
| ☒ | ☐ Eukaryotic cell lines |
| ☒ | ☐ Palaeontology and archaeology |
| ☒ | ☐ Animals and other organisms |
| ☒ | ☐ Clinical data |
| ☒ | ☐ Dual use research of concern |

## Methods

| n/a | Involved in the study |
|-----|----------------------|
| ☒ | ☐ ChIP-seq |
| ☒ | ☐ Flow cytometry |
| ☒ | ☐ MRI-based neuroimaging |

## Antibodies

**Antibodies used**

Antibodies against proteins from baker's yeast Saccharomyces cerevisiae (are custom-made and are not commecially available) were generated in rabbits using peptides (Tim17, Tim21, Tim23, Tim44, Tim50, Mgr2, Aco1, Atp1, Cox1, Cox4, Cox12, Mdh1, Qcr8, Ssc1/mtHsp70, Tim22, Tom5, Tom22, Tom40, Tom70, Yme1). The antisera were used in 1:250-1,000 dilution. Primary anti-GFP Roche (#11814460001, clones 7.1 and 13.1, lot: 65309400) and secondary goat anti-rabbit-HRP Jackson ImmunoResearch Laboratories (#111-035-003, lot: 162282), goat anti-rabbit-HRP Sigma (A6154, lot: SLBG72001V), horse anti-mouse-HRP Cell Signaling Technology (#7076S, lot: 38) antibodies were obtained as listed. Secondary antibodies were used at concentrations of 1:5,000 to 1:20,000 (anti-rabbit HRP) or 1:2,000 (anti-mouse-HRP).

**Validation**

The specificity of the antibody raised against a protein from baker´s yeast (Saccharomyces cerevisiae) was controlled by comparing total cell extracts or mitochondrial lysates from wild-type yeast cells and the corresponding deletion strain or strains expressing a tagged version of the protein of interest via SDS-PAGE and Western blotting. Absence or size shift of the signal in cellular fractions of the mutant strain confirmed the specificity of the antibody signal. References for the antibodies are:
Rabbit polyclonal anti-Tim17, Ref. 65
Rabbit polyclonal anti-Tim21, Ref. 65
Rabbit polyclonal anti-Tim23, Ref. 65
Rabbit polyclonal anti-Tim44, Ref. 65
Rabbit polyclonal anti-Tim50, Ref. 65
Rabbit polyclonal anti-Mgr2, Ref. 65
Rabbit polyclonal anti-Aco1, Ref. 65
Rabbit polyclonal anti-Atp1, Ref. 66
Rabbit polyclonal anti-Cox1, Ref. 65
Rabbit polyclonal anti-Cox4, Ref. 65
Rabbit polyclonal anti-Cox12, Ref. 67
Rabbit polyclonal anti-Mdh1, Ref. 68
Rabbit polyclonal anti-Ssc1/mtHsp70, Ref. 67
Rabbit polyclonal anti-Qcr8, Ref. 65
Rabbit polyclonal anti-Tim22, Ref. 69
Rabbit polyclonal anti-Tom5, Ref. 70
Rabbit polyclonal anti-Tom22, Ref. 65
Rabbit polyclonal anti-Tom40, Ref. 65
Rabbit polyclonal anti-Tom70, Ref. 65
Rabbit polyclonal anti-Yme1, Ref. 65
Mouse monoclonal anti-GFP, Roche, 11814460001, manufacturer tested for functionality and purity relative to a reference standard to confirm the quality of each new reagent preparation, Gomkale et al., 2021, doi:10.1038/s41467-021-26016-1

