## [Peer Review File · Nature]

Manuscript Title: Central role of Tim17 in mitochondrial presequence protein translocation

Reviewer Comments & Author Rebuttals

Reviewer Reports on the Initial Version:

Referees' comments:

Referee #1 (Remarks to the Author):

The manuscript by Fielden et al addresses the role of Tim17, the essential subunit of the TIM23 complex, during translocation of proteins into mitochondria. The TIM23 complex is the major protein translocase of the mitochondrial inner membrane that sorts presequence proteins into the mitochondrial matrix and the inner membrane. Tim17 and Tim23 form the core of the TIM23 complex. The two proteins are homologous to each other, however, they can not substitute for each other. The currently prevailing view is that Tim23 forms a hydrophilic channel for precursor protein translocation and Tim17 has a regulatory function. In the current study, the authors combined structural modelling with biochemical and genetic experiments in yeast to address the function of Tim17. The authors first show that both matrix targeted and inner membrane sorted proteins are in close vicinity to Tim17, however they could not make the same observation for Tim23. Then, they made structural models of Tim17 and Tim23, based on the homology with Tim22 whose structure was recently solved. The generated models suggest that neither Tim17 nor Tim23 form water-filled channels but rather that their 4 transmembrane segments form curved surfaces with the lateral cavities opening to the lipid bilayer. The authors provide evidence that the hydrophilic residues in the Tim17 cavity are crucial for TIM23 structure and/or protein translocation. During this manuscript was in preparation, a structural model of the Tim17-Tim23 core complex was generated based on co-evolution analysis and the preprint describing the cryoEM structure of TIM23 was posted. Both studies showed that Tim17 and Tim23 interact back to back and thus do not form a channel even in a heterodimer. This relative orientation of Tim17 and Tim23, though unexpected, is in agreement with previous mutagenesis analyses and the experiments performed in the current study provide additional support for this arrangement of Tim17 and Tim23. The authors also identify conserved negatively charged residues in Tim17 that are close to the intermembrane space side of the lipid bilayer as essential for protein translocation and note that Tim23 lacks such conserved acidic patch. In the end, the authors propose a provocative model in which proteins are translocated along the Tim17-bilayer interface, the mechanism that would also permit direct lateral release of transmembrane segments of inner membrane sorted proteins and prevent accidental leakage of protons across the inner membrane. The manuscript is well structured and very clearly written. The authors did an excellent job in explaining and also reconciling many previous findings on the chemical features of presequences and in particular on the previously observed channel activities that led to the current view that Tim23 forms a hydrophilic channel. This part of the manuscript is somewhat long compared to the rest but I find it very helpful, especially for nonspecialists. The data are of extraordinary quality, as one would expect from the leaders in the field. The model proposed is similar to the model proposed in the above mentioned preprint describing the cryoEM structure of TIM23. It is provocative and would, in my opinion, need to be supported with more data than

currently presented, see below, but, in general, the manuscript has the potential to appeal to a very broad audience. The points that would need to be addressed in the revised version are listed below.

1. The authors propose a provocative but exciting model which states that the cavity formed by Tim17 represents the translocation path for proteins. The model is based on several observations: a) Tim17 is in contact with both matrix-targeted and inner membrane sorted proteins whereas no such observation could be made for Tim23, b) hydrophilic residues in the Tim17 cavity are important for TIM23 structure and/or protein translocation, and c) negatively charged residues on the intermembrane space side of Tim17 impair cell growth and protein import into mitochondria.

Though these observations support the proposed model, some control experiments are missing.

A) It is in my opinion essential that the authors arrest at least one more matrix-targeted protein, if not also another laterally sorted protein, and confirm the crosslinking pattern they obtained for b2 constructs. This is particularly important as b2DHFR is normally sorted to the inner membrane and though the deletion of its stop-transfer signal leads to the translocation of the protein into the matrix, it can not be excluded that the protein takes the route that is not normally taken by matrix targeted proteins. On page 10, lines 34 to 37, the authors discuss the possibility that “some special precursors might also use the lateral cavity of Tim23” - maybe b2deltaDHFR is a special precursor that uses the lateral cavity of Tim17?

B) The actual gels used for the quantification presented in Fig 1h should be shown in the manuscript, especially since the strongest crosslinks of the matrix-targeted proteins to Tim17 are apparently to the positions not included in Fig 1e.

C) The conclusion that Tim23 is not in contact with arrested proteins is based on a single experiment - arrest of a laterally sorted protein with a photo-activatable lysine residue inserted in one position followed by immunoprecipitation using Tim23 antibodies. This experiment, presented in Ext Data Fig 1a, showed that that no band in the immunoprecipitated material was dependent on the presence of the photo-activatable lysine residue. First, it is unclear why the authors analysed only this one position, especially since numerous studies in the past demonstrated that Tim23 can be crosslinked to arrested precursors. Clearly, a more thorough analysis is needed here. Second, the control that the IP worked at all ie that Tim23 itself was precipitated in this experiment should be shown. Finally, the pattern of this IP looks very similar to the patterns presented in Figs 1c and e and Ext Data Fig 1b that show the IPs of the arrested proteins with Tim17 and Tim21 antibodies. Though the crosslinks to Tim17 and Tim21 were unambiguously identified by shifts seen when tagged versions of Tim17 and Tim21 were used, Tim17, Tim21 and Tim23 all have relatively similar sizes and it would be important to show control IPs with Tim17 and Tim21 antibodies performed with samples in which no photo-activatable lysine residues were present.

D) The data obtained with the mutant in which the hydrophilic residues within the Tim17 cavity were exchanged to bulky hydrophobic leucine residues are, in my opinion, not necessarily supporting the model that Tim17 forms the translocation path for proteins. Namely, the authors show in Fig 2d and state in the text, page 9, lines 8-10, that the triple mutation causes a significant destabilization of the TIM23core complex, raising the possibility that the observed import defects may be a secondary effect of the perturbations of the Tim17-Tim23 interaction and thereby also of the potential translocation path along the Tim23 cavity. A less severe mutant which still impairs import but does not affect Tim17-Tim23 interaction could be designed and analysed. Maybe any of the single or double mutants of the three residues could be tested for this purpose.

E) To identify residues in Tim17 that are in close vicinity to arrested precursors, the authors imported, into isolated mitochondria, 35S-labelled Tim17 molecules with photo-activatable lysine

residues inserted in two different positions followed by arrest of unlabelled precursor protein and IP against the arrested domain. This assay relies on the assumption that the majority, if not all, of the imported Tim17 molecules are correctly assembled with Tim23, which can not be guaranteed. It would be more convincing if a similar approach was taken with photoactivatable residues inserted into Tim17 *in vivo*. Few additional positions should also be chosen in order to cover the entire length of the cavity. Also, labelling of the corresponding positions in Tim23 cavity could provide further support for the model in which Tim17 forms the translocation path for proteins and not Tim23.

2. The disulphide bond formed between cysteine residue introduced in Tim17 and the endogenous one in Tim23 are in agreement with the back to back orientation of Tim17 and Tim23. However, the gels shown in Fig 2f and g demonstrate that this version of Tim17 forms a disulphide bond with some other protein far more efficiently than with Tim23. Choosing additional positions along TMs 1 and 2 of Tim17 and Tim23, similarly to what the authors did in the past for Sam50, porin and Tom40, could provide additional support for this unexpected relative orientation of the two proteins.

3. Many publications (starting from Bauer et al, Cell, 1996 to the very recent Gomkale et al, Nature Communications, 2021) showed that the TIM23 complex contains at least two copies of each Tim17 and Tim23. Since this is not seen in any of the currently available models/structures, it is hard to exclude a possibility that two of the here discussed Tim17-Tim23 heterodimers could come together and form an actual channel (plus two lipid-facing cavities). Also, the structural model which includes Mgr2 suggests that this protein may close the Tim17 cavity potentially forming a channel. Though I personally very much like the idea that there is no water filled channel formed by Tim17-Tim23, this discrepancy should be commented on.

Minor comments:

1. Fig 4 a-g, it is unclear why the authors went through the trouble to generate and then analyse sick Tim17 double mutants in the background of Tom22 shuffling strain in which the chromosomal copy of Tim17 can be downregulated by removal of galactose when already single mutants, in particular D17A, show very prominent import defect.

2. Page 9, lines 3-5, on the argument that the sequence conservation within TMs of Tim17 supports the model in which Tim17 forms the translocation path – one can also argue the opposite: surface exposed residues are more likely to be conserved when they are involved in specific protein-protein interactions rather than when they need to loosely interact with many different translocating proteins.

3. Ext Data Fig 2b, is there a plausible explanation why the Tim17 triple mutant shows a far stronger growth defect on YPD than on YPG?

4. Page 10, line 34 – “to” is missing between “sufficient” and “make”.

5. Page 14, lines 24-28, unclear sentence, consider splitting it into two. Same for the legend of Ext Data Fig 1 on page 24.

6. Page 20, lines 26 and 27, I think *in vitro* or *in organello* would be more appropriate here than *in vivo*.

7. Page 21, line 11, reference is missing.

8. Fig. 1d, “p” and “i” labels should be moved up. Also, what does v stand for in the depiction of b2 crosslink to wt Tim17 on the right?

9. Fig 2c, if b2(167)DHFR was used for import then the mature form is missing.

Referee #2 (Remarks to the Author):

The manuscript presents a very interesting discovery concerning the mechanisms of protein translocation across or in the inner mitochondrial membrane. This is an important topic as several hundreds of proteins use this mechanism to reach their final destination within the compartments of mitochondria. The large body of biochemical data have contributed to the current picture involving the protein players, energetic requirements as well as many mechanistic details of substrate protein transfer through the TIM23 translocase. However, despite enormous progress in structural understanding of other translocases, there has been not much known about the TIM23 translocase. The recent structural development published in BioRxiv has accelerated possible other interpretations of existing data and, more importantly, has allowed to address directly the mechanisms based on structural considerations.

Surprisingly, and in contrast to the majority of current interpretations, all the data presented by the authors point to Tim17 as a subunit of the TIM23 complex that plays a major role in the protein translocation. Using the site-specific crosslinking the authors demonstrated the interactions of Tim17 with the incoming precursors. The negative charges are important for the binding of the presequence/through the positive charges of the entire precursor moiety, and the translocation takes place at the interface and not the in the presumed channel. The new data align very well with the current shift of paradigm concerning the structure and mechanisms of membrane protein insertase discovered earlier and more recently.

The topic is extremely interesting for a broad audience and important, and timely. The paper is well written and the figures are of high quality and generally presented in a clear way.

Although I generally agree with the authors and accept the critical role of Tim17 in the translocation, as well as architectural considerations on the TIM23 complex fully confirming earlier structural data, there are still some issues to be clarified. My major criticism, however, is that the authors seem to make the precise mechanistically and far going conclusions and models on the insufficient basis. I do not feel the data presented on the Tim17 negative charge mutants allow for such broad conclusions, generalizing all the import steps and two major precursors groups, proteins translocated across the membrane and proteins integrated with the inner membrane. In this sense the paper is preliminary. The statements that can be currently made for these mutants are the mutations of these residues keep the TIM23 complex almost intact, at the same time causing the import defect.

- Please provide an evidence that the negative patches of Tim17 are directly involved in the critical interaction with the positive charges of a presequence. This can be done by utilizing site-specific X-links, peptides and/or arrested precursors of defined length.
- Does these Tim17 interactions with a presequence requires the IM potential or perhaps they are stabilized by potential dissipation?
- The temperature sensitive mutants of Tim17, originally published by the Pfanner lab in 2005, and subsequently used by others, have been characterized by unique defects towards the precursor classes. Are these mutants able to engage with a presequence efficiently?
- I am wondering if introducing more negative charges to the cavity of Tim17 would change the dynamics of precursor lateral membrane sorting vs translocation, suggesting that indeed the path, along the cavity as the authors write, is the same for both precursor classes?
- The integrity of the TIM23 complex in these negative charge Tim17 mutants is not fully maintained as shown on the Supplementary Figures. Further characteristics, for example, by the affinity

purification is required.

- Does an engagement of a precursor, one or another kind, with the TIM23 translocase change the back-to-back position as shown by a X-link.
- Are the x-links visible between the precursors and Tim23 (as analyzed in the Fig. 1 for Tim17 and Tim21)
- - The discussion on the translocation mode is very nice, however the model is not justified yet. Hopefully new data will lead to generation of a model, which is less generic (given that it is largely based on the practically publishes paper, currently in BioRXiv)

Referee #3 (Remarks to the Author):

Nuclear-encoded proteins destined for the inner mitochondrial membrane or matrix are synthesized with an N-terminal, positively charged “presequence”. These presequence-containing proteins are ultimately sorted across (or into) the inner membrane by the multisubunit TIM23 complex. Polypeptide translocation is thought to occur through an aqueous channel, but the details have been mysterious. The core transmembrane subunits of the TIM23 complex are two evolutionarily related, four-TMD proteins: Tim23 and Tim17. The function of the Tim17 subunit has been unclear, while Tim23 is thought to form the channel through which clients are moved across or into the membrane.

Here, using a combination of site-specific crosslinking, yeast genetics, mutational analysis, and AlphaFold modeling, the authors define the pathway for presequence translocation. First, the authors identify Tim17 as the primary crosslinked partner of model matrix-targeted and inner membrane sorted preproteins, in native mitochondrial membranes. Second, the authors use AlphaFold to generate a back-to-back heterodimer model of the Tim17:Tim23 complex. Notably, in this arrangement, large, lipid-exposed cavities in each subunit are observed to point away from each other rather than forming a single channel. The relevance of this arrangement is supported by site-specific crosslinking along the predicted dimer interface. It is further supported by a yet-unpublished cryo-EM structure of the Tim17:Tim23 complex that appeared more than a year ago (in 2021) on bioRxiv. Next, the authors use mutational analysis to show that a cluster of conserved, negatively charged residues lining the intermembrane space-side of Tim17 are important for translocation. Finally, the authors use site-specific photocrosslinking to confirm the presequence translocation pathway through the Tim17 cavity.

This is a technically sound paper that proposes new mechanistic insight into translocation of presequence-containing proteins through the TIM23 complex. The most important claim is that this occurs via a lipid-exposed groove presented by Tim17, rather than through a hydrophilic channel defined by Tim23 as previously thought. That Tim17 lines the presequence pathway appears well supported by the author’s data. This is novel, and will be of interest to the protein translocation community. However, it isn’t clear that the data can exclude participation of a functional channel. For example, why do the authors settle on a 1:1 Tim17:Tim23 stoichiometry? Are higher order assemblies possible? This is important because the presented crosslinking and mutant data might also be rationalized by a 2:2 heterotetrameric assembly where the cavities of two Tim17 subunits face each other—thereby providing an aqueous channel for translocation. Alternatively, could

another transmembrane subunit of the TIM23 complex interact with the Tim17 “half-channel” to complete it? These questions might be addressed through AlphaFold modeling that includes additional membrane subunits (e.g. Tim50) and oligomeric states.

Other comments:

-Additional details about the Tim17:Tim23 heterodimer modeling should be provided. In particular, AlphaFold PAE plots for all monomers and complexes, and the resulting models colored by pLDDT (confidence).

-It would be helpful to show hydrophobicity and sequence conservation mapped to the surface of the predicted Tim17:Tim23 heterodimer. Does this help rationalize why Tim17 but not the structurally similar Tim23 groove forms the translocation pathway?

-The abstract and discussion state that the half-channel Tim17 translocation mechanism (as opposed to a full hydrophilic channel) explains how the permeability barrier is maintained. This seems to be an overly simplistic statement since other “half-channel” translocases (including Tim23 and members of the Oxa1 superfamily of insertases) are reported to move ions across the membrane under certain conditions.

-The manuscript is difficult to follow in places. Too many experimental details are presented in the main text (these could be moved to the Methods, figure legends and/or supplement), and the discussion is too long.

Referee #1 (Remarks to the Author):

The manuscript by Fielden *et al* addresses the role of Tim17, the essential subunit of the TIM23 complex, during translocation of proteins into mitochondria. The TIM23 complex is the major protein translocase of the mitochondrial inner membrane that sorts presequence proteins into the mitochondrial matrix and the inner membrane. Tim17 and Tim23 form the core of the TIM23 complex. The two proteins are homologous to each other, however, they can not substitute for each other. The currently prevailing view is that Tim23 forms a hydrophilic channel for precursor protein translocation and Tim17 has a regulatory function. In the current study, the authors combined structural modelling with biochemical and genetic experiments in yeast to address the function of Tim17. The authors first show that both matrix targeted and inner membrane sorted proteins are in close vicinity to Tim17, however they could not make the same observation for Tim23. Then, they made structural models of Tim17 and Tim23, based on the homology with Tim22 whose structure was recently solved. The generated models suggest that neither Tim17 nor Tim23 form water-filled channels but rather that their 4 transmembrane segments form curved surfaces with the lateral cavities opening to the lipid bilayer. The authors provide evidence that the hydrophilic residues in the Tim17 cavity are crucial for TIM23 structure and/or protein translocation. During this manuscript was in preparation, a structural model of the Tim17-Tim23 core complex was generated based on co-evolution analysis and the preprint describing the cryoEM structure of TIM23 was posted. Both studies showed that Tim17 and Tim23 interact back to back and thus do not form a channel even in a heterodimer. This relative orientation of Tim17 and Tim23, though unexpected, is in agreement with previous mutagenesis analyses and the experiments performed in the current study provide additional support for this arrangement of Tim17 and Tim23. The authors also identify conserved negatively charged residues in Tim17 that are close to the intermembrane space side of the lipid bilayer as essential for protein translocation and note that Tim23 lacks such conserved acidic patch. In the end, the authors propose a provocative model in which proteins are translocated along the Tim17-bilayer interface, the mechanism that would also permit direct lateral release of transmembrane segments of inner membrane sorted proteins and prevent accidental leakage of protons across the inner membrane. The manuscript is well structured and very clearly written. The authors did an excellent job in explaining and also reconciling many previous findings on the chemical features of presequences and in particular on the previously observed channel activities that led to the current view that Tim23 forms a hydrophilic channel. This part of the manuscript is somewhat long compared to the rest but I find it very helpful, especially for nonspecialists. The data are of extraordinary quality, as one would expect from the leaders in the field. The model proposed is similar to the model proposed in the above mentioned preprint describing the cryoEM structure of TIM23. It is provocative and would, in my opinion, need to be supported with more data than currently presented, see below, but, in general, the manuscript has the potential to appeal to a very broad audience. The points that would need to be addressed in the revised version are listed below.

1. The authors propose a provocative but exciting model which states that the cavity formed by Tim17 represents the translocation path for proteins. The model is based on several observations: a) Tim17 is in contact with both matrix-targeted and inner membrane sorted proteins whereas no such observation could be made for Tim23, b) hydrophilic residues in the Tim17 cavity are important for TIM23 structure and/or protein translocation, and c) negatively charged residues on the intermembrane space side of Tim17 impair cell growth and protein import into mitochondria. Though these observations support the proposed model, some control experiments are missing.

A) It is in my opinion essential that the authors arrest at least one more matrix-targeted protein, if not also another laterally sorted protein, and confirm the crosslinking pattern they obtained for b2 constructs. This is particularly important as b2DHFR is normally sorted to the inner membrane and though the deletion of its stop-transfer signal leads to the translocation of the protein into the matrix, it can not be excluded that the protein takes the route that is not normally taken by matrix targeted proteins. On page 10, lines 34 to 37, the authors discuss the possibility that "some special precursors might also use the lateral cavity of Tim23" - maybe b2deltaDHFR is a special precursor that uses the lateral cavity of Tim17?

We established a new arrested precursor protein construct Cox5a(1-130)-sfGFP to confirm the crosslinking pattern obtained for the b₂-DHFR constructs (new Extended data Fig. 1d right panel, 8b, e). In summary, we could experimentally show that matrix- and inner membrane-targeted precursor proteins are crosslinked along the lateral Tim17 cavity (new Fig. 1c, 5b-d and new Extended data Fig. 1, 6f, 8b-h), both types of precursor proteins are affected by the mutations of the conserved negatively charged residues located within the lateral cavity of Tim17 (Fig. 3, 4 and Extended data Fig. 7i). Moreover, import of matrix-targeted precursor proteins in the Tim17^{N64L} and Tim17^{S114L} mutants is affected (mutants where hydrophilic residues located within the lateral cavity of Tim17 are replaced by hydrophobic residues, new Fig. 1 e-g and new Extended data Fig. 2). In contrast to Tim17, deletion of negative charges within transmembrane area of Tim23 has no obvious defects (new Extended data Fig. 6g) and introduction of the equivalent negatively charged residues within the lateral cavity of Tim23 cannot rescue the Tim17 mutant lacking the conserved negatively charged

residues (new Extended data Fig. 6h). We are convinced that collectively all these results demonstrate that the Tim17 lateral cavity forms the major translocation path for the membrane potential dependent import of both types of matrix-targeted and inner membrane-sorted mitochondrial precursor proteins with a presequence. Therefore, we are very confident that our conclusions apply to all classical mitochondrial precursor proteins containing a presequence, which are imported in a membrane potential dependent manner across or into the inner membrane. The analysis of special mitochondrial precursor proteins, without classical presequence and/or which have only a very weak or no dependence on the mitochondrial membrane potential was outside the scope of this study. We do not want to exclude the possibility that this very minor fraction of mitochondrial proteins might also employ slightly different mechanisms.

B) The actual gels used for the quantification presented in Fig 1h should be shown in the manuscript, especially since the strongest crosslinks of the matrix-targeted proteins to Tim17 are apparently to the positions not included in Fig 1e. C) The conclusion that Tim23 is not in contact with arrested proteins is based on a single experiment - arrest of a laterally sorted protein with a photo-activable lysine residue inserted in one position followed by immunoprecipitation using Tim23 antibodies. This experiment, presented in Ext Data Fig 1a, showed that that no band in the immunoprecipitated material was dependent on the presence of the photo-activatable lysine residue. First, it is unclear why the authors analysed only this one position, especially since numerous studies in the past demonstrated that Tim23 can be crosslinked to arrested precursors. Clearly, a more thorough analysis is needed here. Second, the control that the IP worked at all ie that Tim23 itself was precipitated in this experiment should be shown. Finally, the pattern of this IP looks very similar to the patterns presented in Figs 1c and e and Ext Data Fig 1b that show the IPs of the arrested proteins with Tim17 and Tim21 antibodies. Though the crosslinks to Tim17 and Tim21 were unambiguously identified by shifts seen when tagged versions of Tim17 and Tim21 were used, Tim17, Tim21 and Tim23 all have relatively similar sizes and it would be important to show control IPs with Tim17 and Tim21 antibodies performed with samples in which no photo-activatable lysine residues were present.

We agree with the referee that it is crucial to analyse if similar crosslinking results can be also obtained for Tim23. Therefore, we have established two new major experimental approaches to specifically distinguish between crosslinking to Tim17 or Tim23: (I) We generated a yeast strain expressing both Tim17 and Tim23 with tags (Tim17^{2xStrep}, ~4 kDa molecular mass shift and His⁵SUMO^{5star}Tim23, ~14 kDa molecular mass shift) to specifically distinguish between Tim17 and Tim23 crosslink formation (new Fig. 1 c, d and Extended data Fig. 1). (II) We generated single cysteine mutants of Tim17^(SCF) and Tim23^(CF) (in the semi cysteine free (SCF) and cysteine free (CF) backgrounds) to perform cysteine-specific crosslinking (new Fig. 5b-d and new Extended data Fig. 6f, 8). In addition, the company (tRNA Probes; <http://www.trnaprobes.com>) which produced the crosslinking reagents required for the photo-crosslinking experiments does not exist anymore and additional experiments using this specific *in vitro* translation approach could not be performed. As we were not able to prove the specificity of all our earlier individual crosslinking bands we therefore replaced the data by new crosslinking approaches where we can specifically distinguish crosslinks formed with Tim17 or Tim23. Our new approaches thus solved the specificity issue to distinguish between Tim17 or Tim23 crosslinks. The new crosslinking experiments demonstrate that the precursor proteins can be specifically crosslinked to residues located along the Tim17 lateral cavity from the intermembrane space side (new Extended data Fig. 6f), the middle of the bilayer, to the matrix side (new Fig. 5b-d and new Extended data Fig. 8 b-e, g, h), while we do not observe similar crosslinks to Tim23 with two different approaches (new Fig. 1c, 5b-d and Extended data Fig. 1, 8c-e, g, h).

D) The data obtained with the mutant in which the hydrophilic residues within the Tim17 cavity were exchanged to bulky hydrophobic leucine residues are, in my opinion, not necessarily supporting the model that Tim17 forms the translocation path for proteins. Namely, the authors show in Fig 2d and state in the text, page 9, lines 8-10, that the triple mutation causes a significant destabilization of the TIM23core complex, raising the possibility that the observed import defects may be a secondary effect of the perturbations of the Tim17-Tim23 interaction and thereby also of the potential translocation path along the Tim23 cavity. A less severe mutant which still impairs import but does not affect Tim17-Tim23 interaction could be designed and analysed. Maybe any of the single or double mutants of the three residues could be tested for this purpose.

Following the suggestion of the referee, we generated new Tim17 mutants where single hydrophilic residues located within the Tim17 lateral cavity were exchanged to hydrophobic leucine residues (new Fig. 1e-g and new Extended data Fig. 2). Isolated Tim17^{N64L} and Tim17^{S114L} mitochondria have comparable protein levels (new Extended data Fig. 2c), assembled TIM23 complexes (new Fig. 1f) and membrane potential (new Extended data Fig. 2d). Import and processing of matrix-targeted precursors is strongly affected (new Fig. 1g), while the processing and assembly of inner membrane-sorted precursors is similar to wildtype (new Extended data Fig. 2e, f). This demonstrates that, in addition to the conserved negative charges, hydrophilic residues situated along the lateral cavity are also crucial for precursor translocation across the inner membrane.

E) To identify residues in Tim17 that are in close vicinity to arrested precursors, the authors imported, into isolated mitochondria, 35S-labelled Tim17 molecules with photo-activatable lysine residues inserted in two different positions followed by arrest of unlabelled precursor protein and IP against the arrested domain. This assay relies on the assumption that the majority, if not all, of the imported Tim17 molecules are correctly assembled with Tim23, which can not be guaranteed. It would be more convincing if a similar approach was taken with photoactivatable residues inserted into Tim17 in vivo. Few additional positions should also be chosen in order to cover the entire length of the cavity. Also, labelling of the corresponding positions in Tim23 cavity could provide further support for the model in which Tim17 forms the translocation path for proteins and not Tim23.

We agree with the referee and therefore have established a new approach. We generated single cysteine mutants of numerous positions within the lateral cavity of Tim17^(SCF) and of Tim23^(CF) (in the semi cysteine free (SCF) and cysteine free (CF) backgrounds). Cysteine-specific crosslinking between arrested precursor proteins and Tim17 specifically along the lateral cavity fully confirm our previous data (new 5b-d and new Extended data Fig. 6f, 8b-h).

2. The disulphide bond formed between cysteine residue introduced in Tim17 and the endogenous one in Tim23 are in agreement with the back to back orientation of Tim17 and Tim23. However, the gels shown in Fig 2f and g demonstrate that this version of Tim17 forms a disulphide bond with some other protein far more efficiently than with Tim23. Choosing additional positions along TMs 1 and 2 of Tim17 and Tim23, similarly to what the authors did in the past for Sam50, porin and Tom40, could provide additional support for this unexpected relative orientation of the two proteins.

We introduced cysteines into the Tim17 semi cysteine free (SCF) and Tim23 cysteine free (CF) background to generate the cysteine pairs Tim17^{F22C} (TM1) with Tim23^{I154C} (TM2) and Tim17^{V76C} (TM2) with Tim23^{L109C} (TM1). In agreement with our previous results we either observe upon oxidation and/or crosslinking, specific disulfide-bonded and/or crosslinked adducts between Tim17 and Tim23 further demonstrating their specific back-to-back orientation (new Extended data Fig. 3f).

3. Many publications (starting from Bauer et al, Cell, 1996 to the very recent Gomkale et al, Nature Communications, 2021) showed that the TIM23 complex contains at least two copies of each Tim17 and Tim23. Since this is not seen in any of the currently available models/structures, it is hard to exclude a possibility that two of the here discussed Tim17-Tim23 heterodimers could come together and form an actual channel (plus two lipid-facing cavities). Also, the structural model which includes Mgr2 suggests that this protein may close the Tim17 cavity potentially forming a channel. Though I personally very much like the idea that there is no water filled channel formed by Tim17-Tim23, this discrepancy should be commented on.

We thank the referee for this important remark. Bauer et al., 1996 Cell (10.1016/S0092-8674(00)81320-3) analysed the potential of Tim23 to dimerize mainly via the N-terminal intermembrane space domain. However, Bauer et al., 1996 Cell already found that 'Tim23 dimers dissociate in a presequence-specific manner'. Gomkale et al., 2021 Nat. Commun. (10.1038/s41467-021-26016-1) co-expressed ALFA-tagged Tim17 or Tim23 in yeast expressing wildtype Tim17 and Tim23 to analyse the oligomerisation state of the translocase. The Tim23^{ALFA} immunoprecipitation (Gomkale et al., Fig. 2c) clearly enriches Tim23^{ALFA} compared to Tim23 in relation to the total. Moreover, while Tim17^{ALFA} is enriched in an extent comparable to Tim23, Tim17^{WT} is extremely reduced in the eluate. Thus, only a minor fraction of Tim17-Tim23 heterodimers associate with each other in the absence of precursor protein. To test the oligomerisation status of the active translocase in the presence of precursor protein, we co-expressed either Tim17^{2xStrep} or HisSUMOstarTim23 in yeast expressing wildtype Tim17 and Tim23. We arrested recombinant Jac1^{sGFP} precursor protein in isolated mitochondria and performed a double purification of the TIM23 complex (1st, 2xStrep-tag via Streptavidin or His-tag via immobilized metal ion affinity chromatography) and of the arrested precursor protein (2nd, sGFP via anti-GFP). In the first purification of the translocase the tagged versions of Tim17 or Tim23 are clearly enriched. Moreover, purification of the TOM-TIM23 supercomplex arrested Jac1^{sGFP} precursor reveals a specific enrichment of Tom40 compared to the isolation of the TIM23 complex, but no further enrichment of the untagged forms of neither Tim17 nor Tim23 (new Fig. 2d, e). These data support the idea, that Tim17-Tim23 heterodimers do not form stable stoichiometric oligomers neither in the absence nor in the presence of arrested precursor protein. This is in agreement with structural modelling of Tim17-Tim23 heterodimers (see also referee #3 general point), which does not support the formation of a specific Tim17-Tim23 channel by heterotetramerisation (the analysis yielded only weak predictions for heterotetrameric arrangements of two Tim17-Tim23 heterotetramers, new Extended data Fig. 4). Taken together, we did not find evidence supporting a Tim17-Tim23 channel like structure by association of 2 lateral cavities. In contrast, the predictions of the Tim17-Mgr2 interaction have a high confidence and support the idea of a partial channel-like structure with hydrophobic interior formed by the lateral cavity of Tim17 and the lateral association of the two α -helices of Mgr2 as predicted by the co-evolutionary analysis published by Humphreys et al., 2021 Science (10.1126/science.abm4805) and depicted in Fig. 6a and new Extended data Fig. 9a.

Minor comments:

1. Fig 4 a-g, it is unclear why the authors went through the trouble to generate and then analyse sick Tim17 double mutants in the background of Tom22 shuffling strain in which the chromosomal copy of Tim17 can be downregulated by removal of galactose when already single mutants, in particular D17A, show very prominent import defect.

We agree with the referee that the Tim17 single negative charge mutants show already prominent import defects. However, to convince referees, editors and readers in a journal with a general audience to change the general dogma of mitochondrial protein translocation across/into the inner membrane we wanted to unambiguously demonstrate, that the charged residues in the lateral cavity of Tim17 are absolutely crucial for mitochondrial protein import.

2. Page 9, lines 3-5, on the argument that the sequence conservation within TMs of Tim17 supports the model in which Tim17 forms the translocation path – one can also argue the opposite: surface exposed residues are more likely to be conserved when they are involved in specific protein-protein interactions rather than when they need to loosely interact with many different translocating proteins.

We thank the referee for this point and therefore removed the corresponding section.

3. Ext Data Fig 2b, is there a plausible explanation why the Tim17 triple mutant shows a far stronger growth defect on YPD than on YPG?

We speculate that the extended doubling time on non-fermentative glycerol medium (YPG) might allow the import of a higher fraction of critical mitochondrial precursor proteins.

4. Page 10, line 34 – “to” is missing between “sufficient” and “make”.

Corrected.

5. Page 14, lines 24-28, unclear sentence, consider splitting it into two. Same for the legend of Ext Data Fig 1 on page 24.

This part of the figure was replaced with new data related to points 1B/C of referee #1.

6. Page 20, lines 26 and 27, I think *in vitro* or *in organello* would be more appropriate here than *in vivo*.

This part of the methods section was replaced to describe the new procedures employed to address points 1B/C of referee #1.

7. Page 21, line 11, reference is missing.

This part of the methods section was replaced to describe the new procedures employed to address points 1B/C of referee #1.

8. Fig. 1d, “p” and “i” labels should be moved up. Also, what does v stand for in the depiction of b2 crosslink to wt Tim17 on the right?

This part of the figure was replaced with new data related to points 1B/C of referee #1.

9. Fig 2c, if b2(167)DHFR was used for import then the mature form is missing.

This part of the figure was replaced with new data related to point 1D of referee #1.

Referee #2 (Remarks to the Author):

The manuscript presents a very interesting discovery concerning the mechanisms of protein translocation across or in the inner mitochondrial membrane. This is an important topic as several hundreds of proteins use this mechanism to reach their final destination within the compartments of mitochondria. The large body of biochemical data have contributed to the current picture involving the protein players, energetic requirements as well as many mechanistic details of substrate protein transfer through the TIM23 translocase. However, despite enormous progress in structural understanding of other translocases, there has been not much known about the TIM23 translocase. The recent structural development published in BioRxiv has accelerated possible other interpretations of existing data and, more importantly, has allowed to address directly the mechanisms based on structural considerations.

Surprisingly, and in contrast to the majority of current interpretations, all the data presented by the authors point to Tim17 as a subunit of the TIM23 complex that plays a major role in the protein translocation. Using the site-specific crosslinking the authors demonstrated the interactions of Tim17 with the incoming precursors. The negative charges are

important for the binding of the presequence/through the positive charges of the entire precursor moiety, and the translocation takes place at the interface and not the in the presumed channel. The new data align very well with the current shift of paradigm concerning the structure and mechanisms of membrane protein insertase discovered earlier and more recently.

The topic is extremely interesting for a broad audience and important, and timely. The paper is well written and the figures are of high quality and generally presented in a clear way.

Although I generally agree with the authors and accept the critical role of Tim17 in the translocation, as well as architectural considerations on the TIM23 complex fully confirming earlier structural data, there are still some issues to be clarified. My major criticism, however, is that the authors seem to make the precise mechanistically and far going conclusions and models on the insufficient basis. I do not feel the data presented on the Tim17 negative charge mutants allow for such broad conclusions, generalizing all the import steps and two major precursors groups, proteins translocated across the membrane and proteins integrated with the inner membrane. In this sense the paper is preliminary. The statements that can be currently made for these mutants are the mutations of these residues keep the TIM23 complex almost intact, at the same time causing the import defect.

- Please provide an evidence that the negative patches of Tim17 are directly involved in the critical interaction with the positive charges of a presequence. This can be done by utilizing site-specific X-links, peptides and/or arrested precursors of defined length.

We thank the referee for the important comment and have generated new yeast strains in the Tim17 semi cysteine free (SCF) background where the conserved negative charges are individually mutated to cysteines. We arrested the matrix-targeted precursor protein $b_2(110)_{\Delta 19}$ -DHFR[#] for cysteine-specific crosslinking with 3-m-maleimidobenzoyl-N-hydroxysuccinimide ester (MBS). The maleimido group has a high specificity for Cys and, in contrast, the N-hydroxysuccinimide-ester has a broad reactivity with Lys, Thr, Ser, Tyr and Arg (Ward et al., 2017 ACS Chem. Biol., 10.1021/acscchembio.7b00125) for crosslinking to the precursor. Both Tim17^{D76C} and Tim17^{E126C} can be specifically crosslinked to an arrested $b_2(110)_{\Delta 19}$ -DHFR[#] precursor (new Extended data Fig. 6f). This demonstrates that the precursor is arrested directly adjacent to the conserved negative charges within the lateral cavity of Tim17.

- Does these Tim17 interactions with a presequence requires the IM potential or perhaps they are stabilized by potential dissipation?

For this analysis we selected the Tim17^{N64C} mitochondria, which had the best crosslinking efficiency to precursor proteins in our analyses and choose the chemical crosslinker 3-m-maleimidobenzoyl-N-hydroxysuccinimide ester (MBS). Membrane potential dependent crosslink formation (new Extended data Fig. 8f) indicated that presequence containing precursor proteins cross the inner membrane in a membrane potential dependent manner at the lateral cavity of Tim17's transmembrane portion.

- The temperature sensitive mutants of Tim17, originally published by the Pfanner lab in 2005, and subsequently used by others, have been characterized by unique defects towards the precursor classes. Are these mutants able to engage with a presequence efficiently?

Following the suggestion of the referee, we performed crosslinking experiments with the temperature sensitive *tim17* mutants. The crosslinking efficiency between arrested matrix-targeted $b_2(110)_{\Delta 19}$ -DHFR or inner membrane-sorted Cox5a(1-130)-sfGFP precursor proteins and the Tim17 temperature sensitive mutants is considerably affected in both mutants (new Extended data Fig. 1d). This indicates that crosslinking of precursor protein to Tim17 directly correlates with the activity of Tim17.

- I am wondering if introducing more negative charges to the cavity of Tim17 would change the dynamics of precursor lateral membrane sorting vs translocation, suggesting that indeed the path, along the cavity as the authors write, is the same for both precursor classes?

We introduced comparably positioned negatively charged residues towards the middle of the Tim17 lateral cavity, ~2 or ~3 turns from the conserved negative charges. Neither Tim17^{M24D_G69D_C118E} nor Tim17^{G28D_F65D_S114E} were viable in the wildtype or Tim17^{D17A_D76A_E126A} background (new Extended data Fig. 6i). In addition, introduction of comparable negative charges within the Tim23 lateral cavity did not rescue the growth defect of the Tim17^{D17A_D76A_E126E} mutant (new Extended data Fig. 6h). These analyses demonstrate that positioning of the conserved negative charges facing the intermembrane space side is crucial to support translocation of mitochondrial presequence proteins across or into the inner membrane.

- The integrity of the TIM23 complex in these negative charge Tim17 mutants is not fully maintained as shown on the

Supplementary Figures. Further characteristics, for example, by the affinity purification is required.

We agree with the referee that negative charge Tim17 double mutant mitochondria isolated from cells grown on YPD/glucose medium at low temperature led to secondary effects on the integrity of the TIM23 complex (as described in the first version of the manuscript). Thus we have removed these subpanels. In the revised manuscript we now point out that the Tim17^{D17A_D76A} mitochondria isolated from the pGAL-TIM17^{WT} depletion in the TOM22Δ background (Fig. 4 and Extended data Fig. 7) show comparable protein levels (Extended data Fig. 7d), TIM23 complex (Fig. 4b and Extended data Fig. 7e, f), membrane potential (Extended data Fig. 7g) and assembly of metabolite carrier proteins (Extended data Fig. 7h) demonstrating the robust integrity of these mutant mitochondria. Functional analyses employing these mutant mitochondria fully support our conclusions on the crucial role of the conserved negative charges within the Tim17 lateral cavity for translocation and membrane insertion of mitochondrial precursor proteins with a presequence.

- Does an engagement of a precursor, one or another kind, with the TIM23 translocase change the back-to-back position as shown by a X-link.

We thank the referee for raising this important point. We now include an experiment where Jac1^{sGFP} precursor is arrested to induce formation of a TOM-TIM23 supercomplex in Tim17^{N16C} and Tim23^{WT} mitochondria (the Tim17-Tim23 heterodimer in back-to-back orientation can be covalently linked by a disulfide bond between Tim17^{N16C} and Tim23^{C98} upon oxidation; Fig. 2b). Analysis of Tim17 and Tim23 isolated by pull-down of the precursor protein revealed that the covalent disulfide bond between Tim17 and Tim23 is formed with similar efficiency by oxidation in the translocation intermediate (new Fig. 2c). Therefore, we conclude that the back-to-back orientation of Tim17 and Tim23 is maintained upon engagement with precursor protein.

- Are the x-links visible between the precursors and Tim23 (as analyzed in the Fig. 1 for Tim17 and Tim21)

With the initial photo-crosslinking approach we were not able to analyse site-specific crosslinking between the precursor and Tim23 because Tim23 already formed non-site specific covalent adducts with the precursor protein lacking the photo-crosslinker (Extended data Fig. 1a first version of the manuscript). Therefore, we now include a new crosslinking approach by introduction of site-specific cysteines into the radiolabelled precursor proteins, import and arrest into WT or Tim17^{2xStrep/HisSUMOstar}Tim23 mitochondria (new Fig. 1c, d and Extended data Fig. 1), followed by cysteine-specific crosslinking. We focused our new analysis on residues between amino acid 47 and 54, which are spanning the inner membrane upon formation of the *b*₂(84)₊₇-DHFR or *b*₂(110)_{Δ19}-DHFR precursor-TOM-TIM23 supercomplexes. We observed consistent crosslinking between the precursor and Tim17 (confirmed by the specific ~4 kDa mass shift of the crosslink bands corresponding to the 2xStrep-tag fused to Tim17) and observe no obvious crosslinking to Tim23 (which should yield a ~14 kDa mass shift with HisSUMOstarTim23). To confirm this analysis, we additionally performed the reverse experiment and introduced *in vivo* specific cysteine residues within the lateral transmembrane facing cavities of Tim17 and Tim23. Upon import and arrest of *b*₂(84)₊₇-DHFR, *b*₂(110)_{Δ19}-DHFR and Cox5a(1-130)-sGFP we observe efficient cysteine-specific crosslinking bands with Tim17 (SCF, semi cysteine free) K36C, N64C, D76C and E126C, but no obvious crosslinks with cysteines specifically introduced at comparable position within the lateral cavity of Tim23 (new Fig. 5b, c and Extended data Fig. 8b-f). To increase the specificity and to reduce the background we combined both approaches and performed a cysteine bispecific crosslinking employing the specific cysteines within the arrested precursor and the specific cysteines within the lateral transmembrane facing cavities of Tim17 and Tim23. This approach also yields specific crosslinks with Tim17 residues K36C, N64C and D76C (new Fig. 5d and new Extended data Fig 8g, h) spanning the Tim17 lateral cavity from the intermembrane space to the matrix side and the efficiency of these crosslinks is influenced by the position of the cysteine within the arrested precursor.

- The discussion on the translocation mode is very nice, however the model is not justified yet. Hopefully new data will lead to generation of a model, which is less generic (given that it is largely based on the practically publishes paper, currently in BioRxiv)

In our revised manuscript we analysed the import of numerous different matrix- and inner membrane-destined precursor proteins. Matrix-targeted and inner membrane-sorted precursor proteins are crosslinked throughout the Tim17 lateral cavity (new Fig. 1c, 5b-d and new Extended data Fig. 1, 6f, 8b-g), are affected by the mutations of Tim17's conserved negatively charged residues located on the intermembrane space side of the lateral cavity (Fig. 3, 4 and Extended data Fig. 7i). Moreover, import of matrix-targeted precursor proteins in the hydrophilic Tim17 cavity mutants N64L and S114L is also affected (new Fig. 1 e-g and new Extended data Fig. 2), deletion of similar negative charges at the cavity of Tim23 causes no obvious defects (new Extended data Fig. 6g) and introduction of the equivalent negative charges within the Tim23 cavity cannot rescue the Tim17 triple negative charge mutant (D17A_D76A_E126A ; new Extended data Fig. 6i). We believe our experiments collectively demonstrate that the Tim17 lateral cavity forms the major translocation path for the membrane potential dependent import of matrix-targeted and inner membrane-sorted mitochondrial presequence proteins.

Referee #3 (Remarks to the Author):

Nuclear-encoded proteins destined for the inner mitochondrial membrane or matrix are synthesized with an N-terminal, positively charged “presequence”. These presequence-containing proteins are ultimately sorted across (or into) the inner membrane by the multisubunit TIM23 complex. Polypeptide translocation is thought to occur through an aqueous channel, but the details have been mysterious. The core transmembrane subunits of the TIM23 complex are two evolutionarily related, four-TMD proteins: Tim23 and Tim17. The function of the Tim17 subunit has been unclear, while Tim23 is thought to form the channel through which clients are moved across or into the membrane.

Here, using a combination of site-specific crosslinking, yeast genetics, mutational analysis, and AlphaFold modeling, the authors define the pathway for presequence translocation. First, the authors identify Tim17 as the primary crosslinked partner of model matrix-targeted and inner membrane sorted preproteins, in native mitochondrial membranes. Second, the authors use AlphaFold to generate a back-to-back heterodimer model of the Tim17:Tim23 complex. Notably, in this arrangement, large, lipid-exposed cavities in each subunit are observed to point away from each other rather than forming a single channel. The relevance of this arrangement is supported by site-specific crosslinking along the predicted dimer interface. It is further supported by a yet-unpublished cryo-EM structure of the Tim17:Tim23 complex that appeared more than a year ago (in 2021) on bioRxiv. Next, the authors use mutational analysis to show that a cluster of conserved, negatively charged residues lining the intermembrane space-side of Tim17 are important for translocation. Finally, the authors use site-specific photocrosslinking to confirm the presequence translocation pathway through the Tim17 cavity.

This is a technically sound paper that proposes new mechanistic insight into translocation of presequence-containing proteins through the TIM23 complex. The most important claim is that this occurs via a lipid-exposed groove presented by Tim17, rather than through a hydrophilic channel defined by Tim23 as previously thought. That Tim17 lines the presequence pathway appears well supported by the author’s data. This is novel, and will be of interest to the protein translocation community. However, it isn’t clear that the data can exclude participation of a functional channel. For example, why do the authors settle on a 1:1 Tim17:Tim23 stoichiometry? Are higher order assemblies possible? This is important because the presented crosslinking and mutant data might also be rationalized by a 2:2 heterotetrameric assembly where the cavities of two Tim17 subunits face each other—thereby providing an aqueous channel for translocation. Alternatively, could another transmembrane subunit of the TIM23 complex interact with the Tim17 “half-channel” to complete it? These questions might be addressed through AlphaFold modeling that includes additional membrane subunits (e.g. Tim50) and oligomeric states.

We thank the referee for this important question. We performed AlphaFold based ColabFold modelling predictions with two Tim17-Tim23 heterodimers. None of the weakly predicted heterotetrameric complexes formed a channel by association of two the lateral cavities, either formed by two Tim17 “half-channels” nor by one Tim17 and one Tim23 “half channel” (the analysis yielded only weak predictions for other heterotetrameric arrangements of two Tim17-Tim23 heterodimers, new Extended data Fig. 4). To address the question of higher order structures of the Tim17-Tim23 heterodimer experimentally, we generated yeast with tagged and untagged copies of Tim17 and Tim23, respectively. After arresting a matrix-destined precursor protein, we purified the Tim17-Tim23 complex by affinity purification of the tagged subunit and observed a strong enrichment of the tagged over the untagged translocase subunit. A second purification of the TOM-TIM23 arrested precursor protein did not change this enrichment of the tagged form of the Tim17-Tim23 heterodimer (new Fig. 2d, e). This demonstrates that the Tim17-Tim23 heterodimer does not form stable oligomers, neither in the absence nor while engaged with precursor protein (see also our reply to point 3 of referee #1). Concerning the question if other transmembrane subunits of the TIM23 complex interact with one of the lateral gates/“half channels”, modelling predicts the direct association of Mgr2 to the lateral gate side of Tim17 in agreement with Humphreys et al., 2021 Science (10.1126/science.abm4805) and depicted in Fig. 6a. Other TIM23 subunits with transmembrane segments Tim50, Tim21 and Pam17 are all predicted to associate at the side of the Tim17 lateral cavity. However, these weaker predictions do not displace Mgr2 and are theoretically compatible with a hexameric assembly of all transmembrane subunits (new Extended data Fig. 9b).

Other comments:

-Additional details about the Tim17:Tim23 heterodimer modeling should be provided. In particular, AlphaFold PAE plots for all monomers and complexes, and the resulting models colored by pLDDT (confidence).

We now include details of the ColabFold modelling and present AlphaFold PAE plots for the Tim17 and Tim23 monomers, the Tim17-Tim23 heterodimer and show the resulting models coloured by pLDDT (new Extended data Fig. 2a, 3a, b, 4, 9b, c).

-It would be helpful to show hydrophobicity and sequence conservation mapped to the surface of the predicted Tim17:Tim23 heterodimer. Does this help rationalize why Tim17 but not the structurally similar Tim23 groove forms the translocation pathway?

We thank the referee for this suggestion. We now show the hydrophobicity and sequence conservation mapped to the surface of the predicted Tim17-Tim23 heterodimer (new Extended data Fig. 3b, 5b, 9a). In contrast to Tim23, the intermembrane space side of the Tim17 lateral cavity forms a large hydrophilic patch, which could be crucial to target the precursor specifically to the entry of the Tim17 lateral cavity (new Extended data Fig. 3b, 9a). In addition, the residues lining the lateral cavity of Tim17 seem to be much more conserved (new Extended data Fig. 5b). Therefore, both analyses support our model that mitochondrial precursor proteins with a presequence are translocated along the lateral cavity of Tim17.

-The abstract and discussion state that the half-channel Tim17 translocation mechanism (as opposed to a full hydrophilic channel) explains how the permeability barrier is maintained. This seems to be an overly simplistic statement since other "half-channel" translocases (including Tim23 and members of the Oxa1 superfamily of insertases) are reported to move ions across the membrane under certain conditions.

We agree with the reviewer and therefore removed these statements.

-The manuscript is difficult to follow in places. Too many experimental details are presented in the main text (these could be moved to the Methods, figure legends and/or supplement), and the discussion is too long.

We revised the main text of our manuscript and moved numerous experimental details to the materials and figure legends, methods and extended data sections to make the manuscript more appealing for a general audience.

Reviewer Reports on the First Revision:

Referees' comments:

Referee #1 (Remarks to the Author):

The authors have addressed the comments raised in a satisfactory manner.

Referee #2 (Remarks to the Author):

I am fully satisfied with the revisions by the authors. This is an exciting and fully demonstrated development in the field.

Referee #3 (Remarks to the Author):

The reviewers have satisfactorily addressed my concerns, and the manuscript is improved. I am happy to recommend it for publication.